# Regulatory T cells suppress the formation of potent KLRK1 and IL-7R expressing effector CD8 T cells by limiting IL-2

Oksana Tsyklauri[1,2], Tereza Chadimova[1,2], Veronika Niederlova[1,2], Jirina Kovarova[3], Juraj Michalik[1], Iva Malatova[3], Sarka Janusova[1], Olha Ivashchenko[1], Helene Rossez[4], Ales Drobek[1], Hana Vecerova[1], Virginie Galati[4], Marek Kovar[3], Ondrej Stepanek[1,4]*

[1]Laboratory of Adaptive Immunity, Institute of Molecular Genetics of the Czech Academy of Sciences, Prague, Czech Republic; [2]Faculty of Science, Charles University, Prague, Czech Republic; [3]Institute of Microbiology of the Czech Academy of Sciences, Prague, Czech Republic; [4]Department of Biomedicine, University Hospital of Basel, Basel, Switzerland

**Abstract** Regulatory T cells (Tregs) are indispensable for maintaining self-tolerance by suppressing conventional T cells. On the other hand, Tregs promote tumor growth by inhibiting anti-cancer immunity. In this study, we identified that Tregs increase the quorum of self-reactive CD8[+] T cells required for the induction of experimental autoimmune diabetes in mice. Their major suppression mechanism is limiting available IL-2, an essential T-cell cytokine. Specifically, Tregs inhibit the formation of a previously uncharacterized subset of antigen-stimulated KLRK1[+] IL-7R[+] (KILR) CD8[+] effector T cells, which are distinct from conventional effector CD8[+] T cells. KILR CD8[+] T cells show superior cell-killing abilities in vivo. The administration of agonistic IL-2 immunocomplexes phenocopies the absence of Tregs, i.e., it induces KILR CD8[+] T cells, promotes autoimmunity, and enhances antitumor responses in mice. Counterparts of KILR CD8[+] T cells were found in the human blood, revealing them as a potential target for immunotherapy.

## Editor's evaluation

This important study extends our understanding of how regulatory T cells can modulate the function of effector CD8 T cells. The evidence supporting the claims of the authors is solid and direct test the proposed mode of action. The work will be of interest to immunologists working on immune cell regulation.

*For correspondence:
ondrej.stepanek@img.cas.cz

Competing interest: The authors declare that no competing interests exist.

## Introduction

Physiological immune responses aim at invading pathogens, but not at healthy tissues. There are two major principles of controlling self-reactive lymphocytes: clonal deletion (central tolerance) and suppression (peripheral tolerance). FOXP3[+] regulatory T cells (Tregs) represent a major force of peripheral tolerance as they regulate homeostasis and immune responses of conventional T cells (*Josefowicz and Rudensky, 2009*). The absence of Tregs in FOXP3-deficient individuals leads to a severe autoimmune condition called immunodysregulation polyendocrinopathy enteropathy X-linked syndrome (*Bennett et al., 2001*; *Brunkow et al., 2001*; *Wildin et al., 2001*). On the other hand, Tregs inhibit antitumor immune responses (*Togashi et al., 2019*).

**eLife digest** As well as protecting us from invading pathogens, like bacteria or viruses, our immune system can also identify dangerous cells of our own that may cause the body harm, such as cancer cells. Once detected, a population of immune cells called cytotoxic T cells launch into action to kill the potentially harmful cell. However, sometimes the immune system makes mistakes and attacks healthy cells which it misidentifies as being dangerous, leading to autoimmune diseases.

Special immune cells called T regulatory lymphocytes, or 'Tregs', can suppress the activity of cytotoxic T cells, preventing them from hurting the body's own cells. While this can have a positive impact and reduce the effects of autoimmunity, Tregs can also make the immune system less responsive to cancer cells and allow tumors to grow. But how Tregs alter the behavior of cytotoxic T cells during autoimmune diseases and cancer is poorly understood. While multiple mechanisms have been proposed, none of these have been tested in living animal models of these diseases.

To address this, Tsyklauri et al. studied Tregs in laboratory mice which had been modified to have autoimmune diabetes, which is when the body attacks the cells responsible for producing insulin. The experiments revealed that Tregs take up a critical signaling molecule called IL-2 which cytotoxic T cells need to survive and multiply. As a result, there is less IL-2 molecules available in the environment, inhibiting the cytotoxic T cells' activity. Furthermore, if Tregs are absent and there is an excess of IL-2, this causes cytotoxic T cells to transition into a previously unknown subset of T cells with superior killing abilities.

Tsyklauri et al. were able to replicate these findings in two different groups of laboratory mice which had been modified to have cancer. This suggests that Tregs suppress the immune response to cancer cells and prevent autoimmunity using the same mechanism. In the future, this work could help researchers to develop therapies that alter the behavior of cytotoxic T cells and/or Tregs to either counteract autoimmune diseases, or help the body fight off cancer.

---

Although most studies have focused on how Tregs suppress conventional CD4[+] T cells, CD8[+] T cells are also regulated by Tregs. At steady state, Tregs suppress the proliferation of CD8[+] T cells (***Chinen et al., 2016***) and prevent the spontaneous differentiation of memory CD8[+] T cells into effector cells (***Laidlaw et al., 2015***; ***Kalia et al., 2015***). Moreover, Tregs suppress antigenic responses of CD8[+] T cells (***McNally et al., 2011***; ***Kastenmuller et al., 2011***; ***Pace et al., 2012***). Multiple mechanisms of Treg-mediated suppression of CD8[+] T cells have been proposed, such as reducing the expression of co-stimulatory molecules on antigen-presenting cells (APC) via CTLA4 (***Kalia et al., 2015***; ***Kastenmuller et al., 2011***; ***Schildknecht et al., 2010***), limiting the availability of IL-2 (***McNally et al., 2011***; ***Kastenmuller et al., 2011***), production of anti-inflammatory cytokines IL-10 (***Laidlaw et al., 2015***) and TGFβ (***Green et al., 2003***), and limiting the production of CCL3, CCL4, and CCL5 chemokines by APCs (***Pace et al., 2012***). However, none of these studies investigated the interaction between Tregs and CD8[+] T cells in an autoimmune model. Thus, it is still unclear which of these mechanisms is the most important for preventing self-reactive CD8[+] T cells from inducing an autoimmune disease.

A single study addressed how Tregs suppress CD8[+] T-cell responses with various affinities to the cognate antigen (***Pace et al., 2012***). The authors concluded that Tregs inhibit CD8[+] T-cell activation by low-affinity, but not high-affinity, antigens. In such a case, Tregs would not correct errors of central tolerance by suppressing highly self-reactive CD8[+] T cells that escape negative selection in the thymus. Their tolerogenic role would be limited to increasing the antigen-affinity threshold in the periphery (***King et al., 2012***) by suppressing positively selected CD8[+] T cells with intermediate to low affinity to self-antigens.

In this study, we focused on Treg-mediated peripheral tolerance, which prevents self-reactive CD8[+] T cells from inducing an autoimmune pathology. We found that Tregs increase the number of self-reactive CD8[+] T cells required to trigger experimental diabetes across a wide range of their affinities to self-antigens. In addition, we observed that excessive IL-2 prevents Tregs from suppressing CD8[+] T cells in vitro as well as during autoimmune and antitumor responses in vivo. Moreover, Treg depletion, as well as high IL-2 levels, induces the expansion of an unusual population of potent cytotoxic KLRK1[+] IL-7R[+] CD8[+] T cells. Altogether, our data provide strong evidence that the major mechanism of Treg suppression of CD8[+] T cells is limiting the availability of IL-2.

## Results

### Tregs increase the quorum of self-reactive CD8[+] T cells for inducing an autoimmune pathology

To study the potential role of FOXP3[+] Tregs in the autoimmune response of self-reactive T cells, we employed a model of experimental autoimmune diabetes based on the transfer of ovalbumin (OVA)-specific OT-I T cells into double-transgenic mice expressing ovalbumin in pancreatic β-cells (*Kurts et al., 1998*) and diphtheria toxin receptor (DTR) in FOXP3[+] Tregs. Most of the experiments were performed using RIP.OVA DEREG mice carrying a random insertion of a transgene encoding for DTR-GFP fusion protein under the control of *Foxp3* promoter (*Lahl et al., 2007*), but some experiments were performed using *Foxp3^DTR* RIP.OVA strain with DTR-GFP knocked-in into the 3' untranslated region of the *Foxp3* locus (*Kim et al., 2007*). Subsequent priming of OT-I T cells induced the formation of effector T cells and eventual destruction of pancreatic β-cells. Administration of diphtheria toxin (DT) selectively depleted Tregs uncovering their role in this autoimmune pathology. The major advantages of this model are (i) known etiology – the pathology is triggered by self-reactive CD8[+] T cells, (ii) control of key parameters (number of transferred OT-I T cells, affinity of the priming antigen, presence of Tregs, etc.), and (iii) the bypass of central tolerance, allowing us to study the mechanisms of peripheral tolerance separately from the central tolerance.

The induction of diabetes in the majority of Treg-replete mice required $10^6$ adoptively transferred OT-I T cells followed by priming with OVA peptide and LPS (*Figure 1A–C*). This revealed efficient mechanisms of peripheral tolerance preventing the self-reactive T cells from destroying the insulin-producing cells. After the depletion of Tregs, the number of transferred OT-I T cells, sufficient to induce diabetes in the vast majority of animals, dropped dramatically (*Figure 1B and C*, *Figure 1—figure supplement 1A*). We observed that increased glucose levels in the urine and blood are preceded by abnormal morphology of the pancreatic islets and loss of insulin production together with the infiltration of CD8[+] and CD4[+] T cells in the islets on day 4 post immunization (*Figure 1D and E*). Accordingly, the expansion of OT-I T cells in the spleen was enhanced in Treg-deficient mice compared with Treg-replete mice (*Figure 1F*, *Figure 1—figure supplement 1B and C*). When $10^6$ OT-I T cells were transferred, 50% of Treg-depleted mice developed diabetes, even though they were primed with LPS alone without the cognate peptide (*Figure 1G*, *Figure 1—figure supplement 1D*). This was possible because the small amount of endogenous OVA was sufficient to prime OT-I T cells in the absence of Tregs, as revealed by more robust expansion and CD44[+] CD62L[-] effector T-cell formation in *Foxp3^DTR* RIP.OVA than in *Foxp3^DTR* mice (*Figure 1H*, *Figure 1—figure supplement 1E*). Overall, these results revealed that Tregs represent an important mechanism of peripheral tolerance that prevents the induction of diabetes by self-reactive CD8[+] T cells.

To investigate how Tregs suppress priming of self-reactive T cells by antigens with various affinities, we applied a priming protocol based on the adoptive transfer of bone marrow-derived dendritic cells (DCs) loaded with OVA peptide ($K_D \sim 50$ μM) or its variants recognized by OT-I T cells with lower affinity; Q4R7 ($K_D \sim 300$ μM) or Q4H7 ($K_D \sim 850$ μM) (*Stepanek et al., 2014*). In contrast to OVA peptide and LPS priming, as few as $10^3$ transferred OT-I T cells were able to induce autoimmune diabetes in RIP.OVA mice upon DC-OVA priming (*Figure 2A*). When $3 \times 10^2$ OT-I T cells were transferred, all Treg-depleted mice, but only one third of Treg-replete mice, were diabetic (*Figure 2A*). Accordingly, the absence of Tregs increased the susceptibility to diabetes upon priming with DC-Q4R7, although as many as $3 \times 10^4$ OT-I T cells were required to induce diabetes in most Treg-depleted hosts (*Figure 2B*). Along this line, Treg-replete RIP.OVA mice were resistant to $10^6$ OT-I T cells primed with DC-Q4H7, whereas most Treg-depleted mice manifested diabetes (*Figure 2C*). The expansion of OT-I T cells, their expression of IL-2Rα (CD25; a subunit of the high-affinity IL-2 receptor) and KLRG1 (a marker of short-lived effector cells), and the absolute number of KLRG1[+] OT-I T cells in the spleen were greater in Treg-depleted mice than in Treg-replete RIP.OVA mice on day 6 post-immunization with DC-OVA, -Q4R7, or -Q4H7 (*Figure 2D–F*, *Figure 2—figure supplement 1A–C*). Please note that different numbers of OT-I cells were transferred for immunizations with different peptides in these experiments. DCs not loaded with any peptide did not induce diabetes with one exceptional mouse (*Figure 2—figure supplement 1D and E*). When we titrated the amount of transferred OT-I T cells, we observed that the hierarchy of the biological potencies of high-, intermediate-, and low-affinity antigens was preserved in Treg-deficient and Treg-replete hosts (*Figure 2—figure supplement 1F*). Collectively, these results showed that Tregs increase the

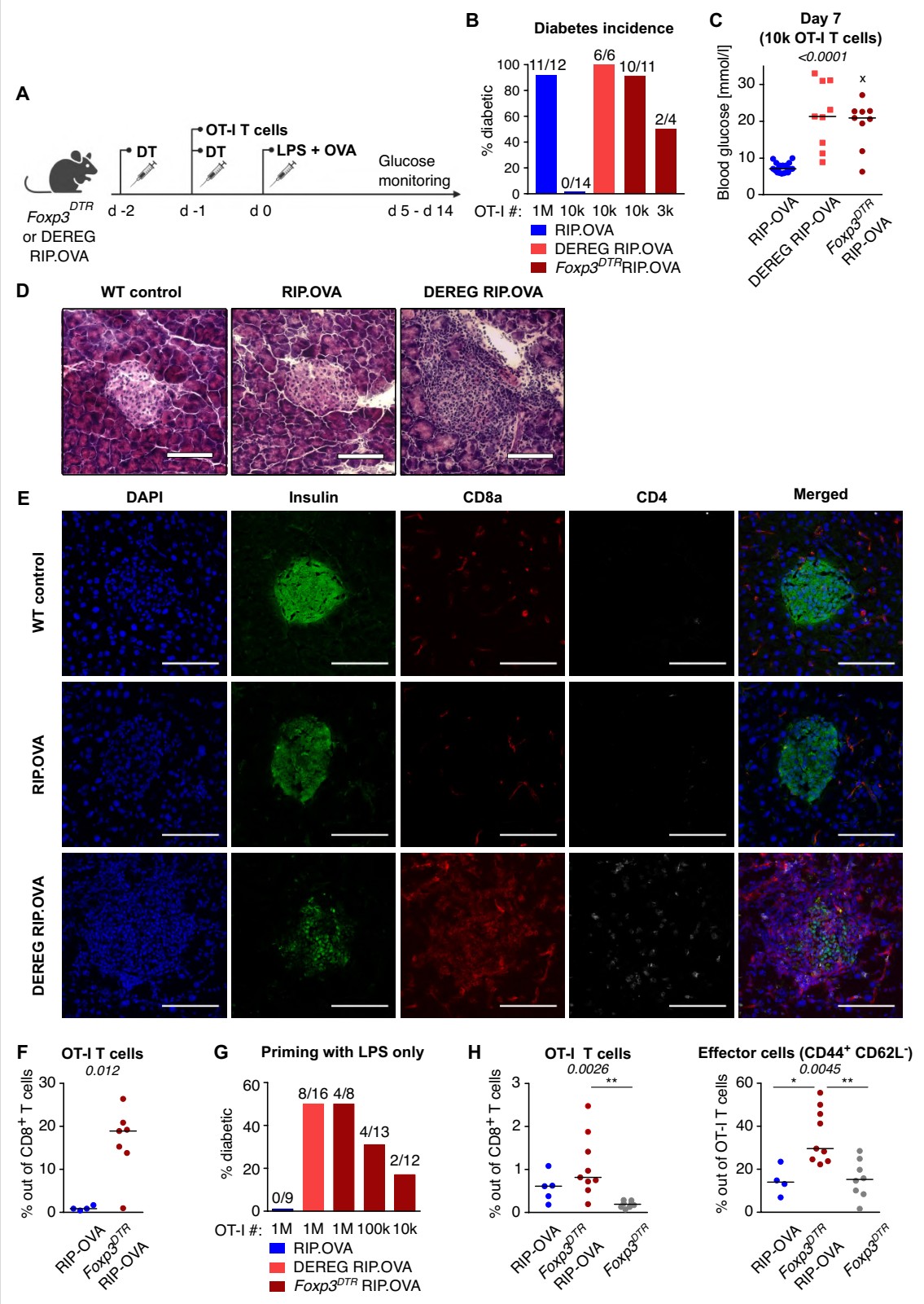

**Figure 1.** Depletion of Tregs decreases the quorum of self-reactive CD8+ T cells required for diabetes induction. (**A**) Scheme of RIP. OVA diabetes model. OT-I T cells were transferred into *Foxp3^DTR* RIP.OVA or DEREG RIP.OVA mice (Treg-depleted with DT) and RIP.OVA controls (Treg-replete). The next day, mice were immunized with OVA peptide and LPS. Urine glucose levels were monitored on a daily basis for 14 days. (**B–H**) Diabetes was induced in RIP.OVA mice as described in (**A**). (**B**) Percentage of diabetic mice is shown. Number of diabetic mice and total number of mice per

*Figure 1 continued on next page*

*Figure 1 continued*

group is indicated on top of each column. Number of transferred OT-I T cells is indicated for each group. (**C**) Glucose concentration in blood on day 7 post-immunization is shown. RIP.OVA n = 19, DEREG RIP.OVA n = 9, *Foxp3^DTR* RIP.OVA n = 9. One mouse from *Foxp3^DTR* RIP.OVA group died before the measurement (shown as 'x'). (**D, E**) Histological analysis of pancreas tissue sections from C57Bl/6J (n = 2), RIP.OVA (n = 4), and DEREG RIP.OVA (n = 4) mice on a day 4 post immunization. Representative images are shown. Two independent experiments were performed. Scale bar 200 µm. (**D**) Hematoxylin and eosin staining. A pancreatic islets is in the center of each image. (**E**) Immunofluorescence staining of nuclei (DAPI) and indicated markers with antibodies. (**F**) Diabetes was induced in RIP.OVA (Ly5.1/Ly5.2, n = 4) and *Foxp3^DTR* RIP.OVA (Ly5.1, n = 7) mice using 0.5 × 10^6 OT-I T cells. Spleens were collected on day 5 and analyzed by flow cytometry. Percentage of OT-I T cells in CD8^+ T-cell population is shown. (**G, H**) RIP.OVA (Ly5.1/Ly5.2), *Foxp3^DTR* RIP.OVA (Ly5.1), DEREG RIP.OVA (Ly5.1/Ly5.2), and *Foxp3^DTR* (Ly5.1) mice were treated as shown in (**A**), with the exception that on day 0, mice were stimulated with LPS only (without OVA peptide). (**G**) Percentage of diabetic mice is shown. Number of diabetic mice and total number of mice per group is indicated on top of each column. Number of transferred OT-I T cells is indicated for each group. (**H**) Diabetes was induced using 0.5 × 10^6 OT-I T cells. Spleens were collected on day 5 and analyzed by flow cytometry. Left: percentage of OT-I T cells in CD8^+ T-cell population is shown. RIP.OVA n = 5, *Foxp3^DTR* RIP.OVA n = 9, *Foxp3^DTR* n = 8. Right: percentage of effector cells defined as CD44^+ CD62L^- in OT-I T-cell population is shown. RIP.OVA n = 4, *Foxp3^DTR* RIP.OVA n = 9, *Foxp3^DTR* n = 8. Statistical significance was calculated by Kruskal–Wallis test (p-value is shown in italics) with Dunn's post-test (*<0.05, **<0.01) for comparison of three groups (**C, F**), or two-tailed Mann–Whitney test for comparison of two groups (**D**, p-value shown in italics). Median is shown.

The online version of this article includes the following figure supplement(s) for figure 1:

**Figure supplement 1.** Suppression of self-reactive CD8^+ T cells by Tregs.

quorum of self-reactive CD8^+ T cells required for inducing autoimmune diabetes, irrespective of the affinity of the priming antigen.

To elucidate, whether the physiological activity of Tregs is a limiting factor of self-tolerance, we selectively expanded the Treg compartment prior to the diabetes induction using IL-2 and anti-IL-2 mAb JES6-1A12 (JES6) immunocomplexes (IL-2ic) (*Polhill et al., 2012*; *Liu et al., 2010*; *Figure 2G*). We observed that the expanded Tregs prevented DC-OVA-induced diabetes in most animals (*Figure 2H*), showing that enhancing the overall Treg activity increases self-tolerance and prevents the induction of the experimental diabetes.

## Tregs suppress self-reactive CD8^+ T cells in the absence of conventional CD4^+ T cells

Tregs could inhibit self-reactive CD8^+ T cells directly or via the suppressing a bystander help of CD4^+ T cells. To address this question, we used T-cell-deficient *Cd3e^-/-* RIP.OVA mice as hosts. First, we adoptively transferred 10^6 polyclonal OVA-tolerant CD8^+ T cells isolated from RIP.OVA mice to replenish the CD8^+ T-cell compartment in the *Cd3e^-/-* RIP.OVA mice. One week later, we transferred conventional (GFP^-) or Treg (GFP^+) CD4^+ T cells from DEREG RIP.OVA mice. A control group of mice did not receive any CD4^+ T cells. Finally, we transferred OT-I T cells and subsequently primed them with DC-OVA (*Figure 3A*). We observed that conventional CD4^+ T cells, albeit presumably tolerant to OVA, increased the incidence of experimental diabetes and accelerated its onset (*Figure 3B and C*, *Figure 3—figure supplement 1A*). In contrast, Tregs reduced the incidence of diabetes compared with mice receiving no CD4^+ T cells (*Figure 3B and C*, *Figure 3—figure supplement 1A*). Conventional CD4^+ T cells enhanced the expansion of OT-I T cells and increased the number of KLRG1^+ OT-I T cells, whereas Tregs elicited the opposite effect (*Figure 3D and E*, *Figure 3—figure supplement 1B and C*), as revealed on day 8 post-immunization. The role of Tregs was even more apparent on day 14 post-immunization, which corresponded to the typical onset of diabetes in the absence of conventional CD4^+ T cells (*Figure 3F and G*, *Figure 3—figure supplement 1D and E*).

Overall, these experiments showed that conventional CD4^+ T cells provide a bystander help to self-reactive CD8^+ T cells, whereas Tregs directly suppress the priming of CD8^+ T cells.

## Tregs suppress self-reactive CD8^+ T cells via limiting IL-2

One of the Treg-mediated effects observed in the previous experiments was the downregulation of IL-2Rα in CD8^+ T cells. IL-2Rα expression on T cells can be induced by antigen stimulation or by IL-2 itself in a positive feedback loop (*Sereti et al., 2000*). Accordingly, downregulation of IL-2Rα in OT-I T cells in the presence of Tregs might be a consequence of the limited availability of IL-2. We observed significantly higher levels of IL-2Rα and an IL-2 signaling intermediate, pSTAT5 in OT-I T cells in Treg-deficient mice early after priming (*Figure 4A and B*, *Figure 4—figure supplement 1A and B*). These

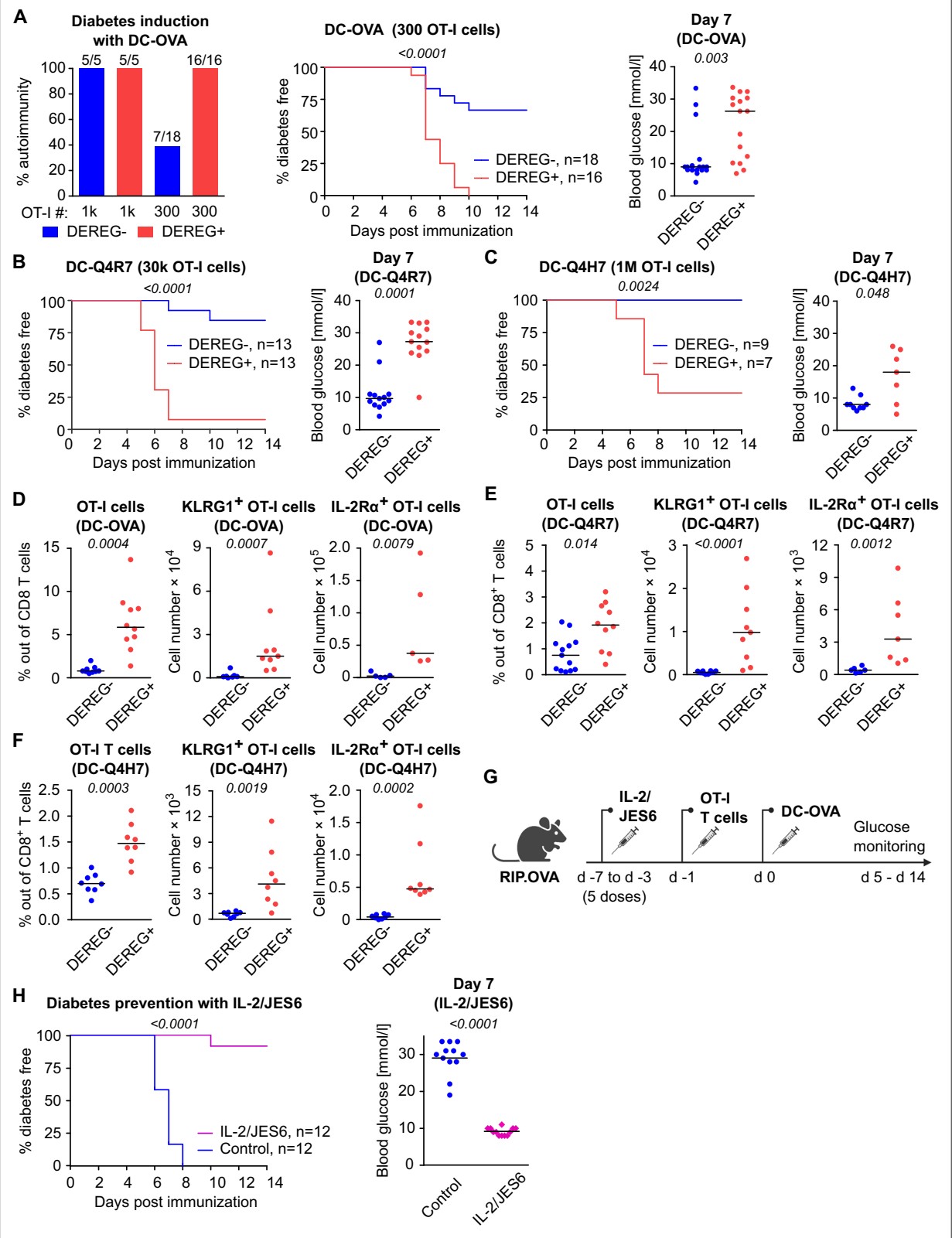

**Figure 2.** Both high-affinity and low-affinity self-reactive CD8[+] T cells are controlled by Tregs. (**A–C**) Treg-depleted DEREG[+] RIP.OVA mice and control DEREG[-] RIP.OVA mice received indicated numbers of OT-I T cells ($10^3$ or $3 \times 10^2$ OT-I T cells in **A**, $3 \times 10^4$ OT-I T cells in **B**, $10^6$ OT-I T cells in **C**). The next day, mice were immunized with DC loaded with an indicated peptide (OVA in **A**, Q4R7 in **B**, Q4H7 in **C**). Urine glucose level was monitored on a daily basis for 14 days. (**A**) Left: percentage of diabetic mice is shown. Number of diabetic mice and total number of mice per group is indicated on

*Figure 2 continued on next page*

*Figure 2 continued*

top of each column. Middle: survival curve. Number of mice per group is indicated. Right: blood glucose concentration on day 7 post-immunization. (**B, C**) Left: survival curve. Number of mice per group is indicated. Right: blood glucose concentration on day 7 post-immunization. (**D–F**) Diabetes was induced in DEREG- RIP.OVA and DEREG+ RIP.OVA mice similarly to (**A–C**). On day 6, spleens were collected and analyzed by flow cytometry. Percentage of OT-I T cells among CD8+ T cells, count of KLRG1+ OT-I T cells, and count of IL-2Rα+ OT-I T cells are shown. (**D**) Diabetes was induced using DC-OVA and $10^3$ OT-I T cells. Left: DEREG- n = 9, DEREG+ n = 10. Middle: DEREG- n = 7, DEREG+ n = 9. Right: n = 5 mice per group. (**E**) Diabetes was induced using DC-Q4R7 and $3 \times 10^4$ OT-I T cells. Left: DEREG- n = 13, DEREG+ n = 10. Middle: DEREG- n = 10, DEREG+ n = 9. Right: DEREG- n = 6, DEREG+ n = 7. (**F**) Diabetes was induced using DC-Q4H7 and $10^6$ OT-I T cells, n = 8 mice per group. (**G, H**) RIP.OVA mice were treated or not with IL-2/JES6 for five consecutive days. Two days after the last dose, the mice received $10^3$ OT-I T cells, and the next day they were immunized with DC-OVA. Urine glucose level was monitored on a daily basis for 14 days. (**G**) Experimental scheme. (**H**) Left: survival curve. Right: blood glucose concentration on day 7 post-immunization. n = 12 mice per group. Statistical significance was calculated by log-rank (Mantel–Cox) test (survival) or two-tailed Mann–Whitney test (glucose concentration and flow cytometry analysis). p-value is shown in italics. Median is shown.

The online version of this article includes the following figure supplement(s) for figure 2:

**Figure supplement 1.** Treg-mediated suppression of CD8+ T cells with various affinities to the antigen.

results were reproduced when congenic Ly5.1 OT-I T cells were used (*Figure 4—figure supplement 1C–E*). These results showed that Tregs limit IL-2 signaling in self-reactive CD8+ T cells in vivo.

Tregs efficiently suppressed proliferation and IL-2Rα expression in CD8+ T cells ex vivo (*Figure 4C*). An addition of excessive IL-2 abolished this suppression (*Figure 4C*). To address whether Tregs can suppress self-reactive CD8+ T cells in the excess of IL-2 in vivo, we performed the diabetic assay with

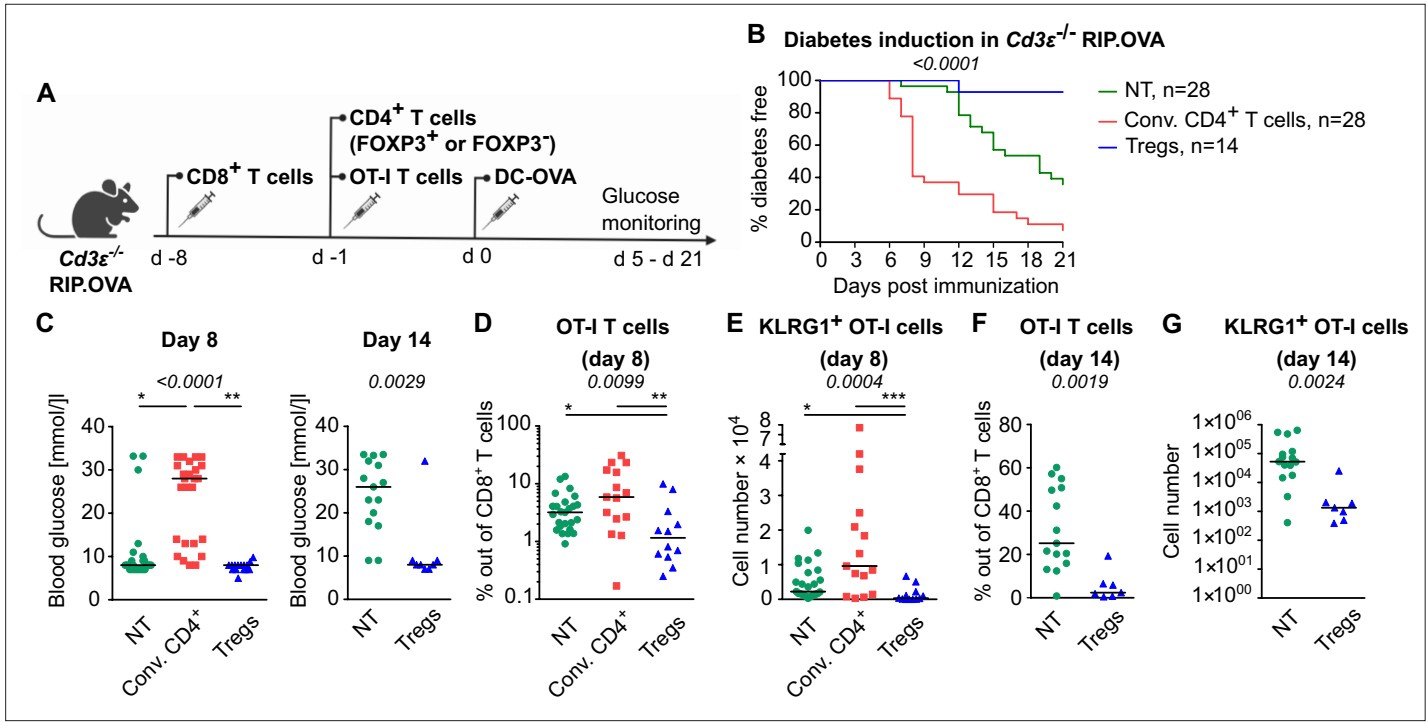

**Figure 3.** Tregs suppress self-reactive CD8+ T cells in the absence of conventional CD4+ T cells. *Cd3e-/-* RIP.OVA mice received $10^6$ polyclonal CD8+ T cells from Ly5.1 RIP.OVA mice. After 7 days, they received either $10^6$ conventional CD4+ T cells (GFP-), or $0.4–1 \times 10^6$ Tregs (GFP+) from DEREG+ RIP. OVA mice, or no CD4+ T cell transfer (NT). Next, 250 OT-I T cells were adoptively transferred to all the recipients. Next day, mice were immunized with DC-OVA. (**A**) Experimental scheme. (**B, C**) Urine glucose level was monitored on a daily basis until day 21 post-immunization. (**B**) Survival curve. Number of mice is indicated. (**C**) Blood glucose concentration on day 8 and day 14 post-immunization is shown. Day 8: NT n = 28, Conv.CD4+ n = 25, Tregs n = 15. Day 14: NT n = 15, Tregs n = 8. (**D–G**) On day 8 or day 14, spleens were collected and analyzed by flow cytometry. Percentage of OT-I T cells among CD8+ T cells (**D, F**), and count of KLRG1+ OT-I T cells (**E, G**) are shown. Day 8: NT n = 25, Conv.CD4+ n = 15, Tregs n = 12. Day 14: NT n = 15, Tregs n = 7. Statistical significance was calculated by Kruskal–Wallis test (p-value is shown in italics) with Dunn's post-test (*<0.05, **<0.01, ***<0.001) for comparison of three groups, or two-tailed Mann–Whitney test for comparison of two groups (p-value is shown in italics). Median is shown.

The online version of this article includes the following figure supplement(s) for figure 3:

**Figure supplement 1.** Suppression of CD8+ T cells by Tregs in the absence of conventional CD4+ T cells.

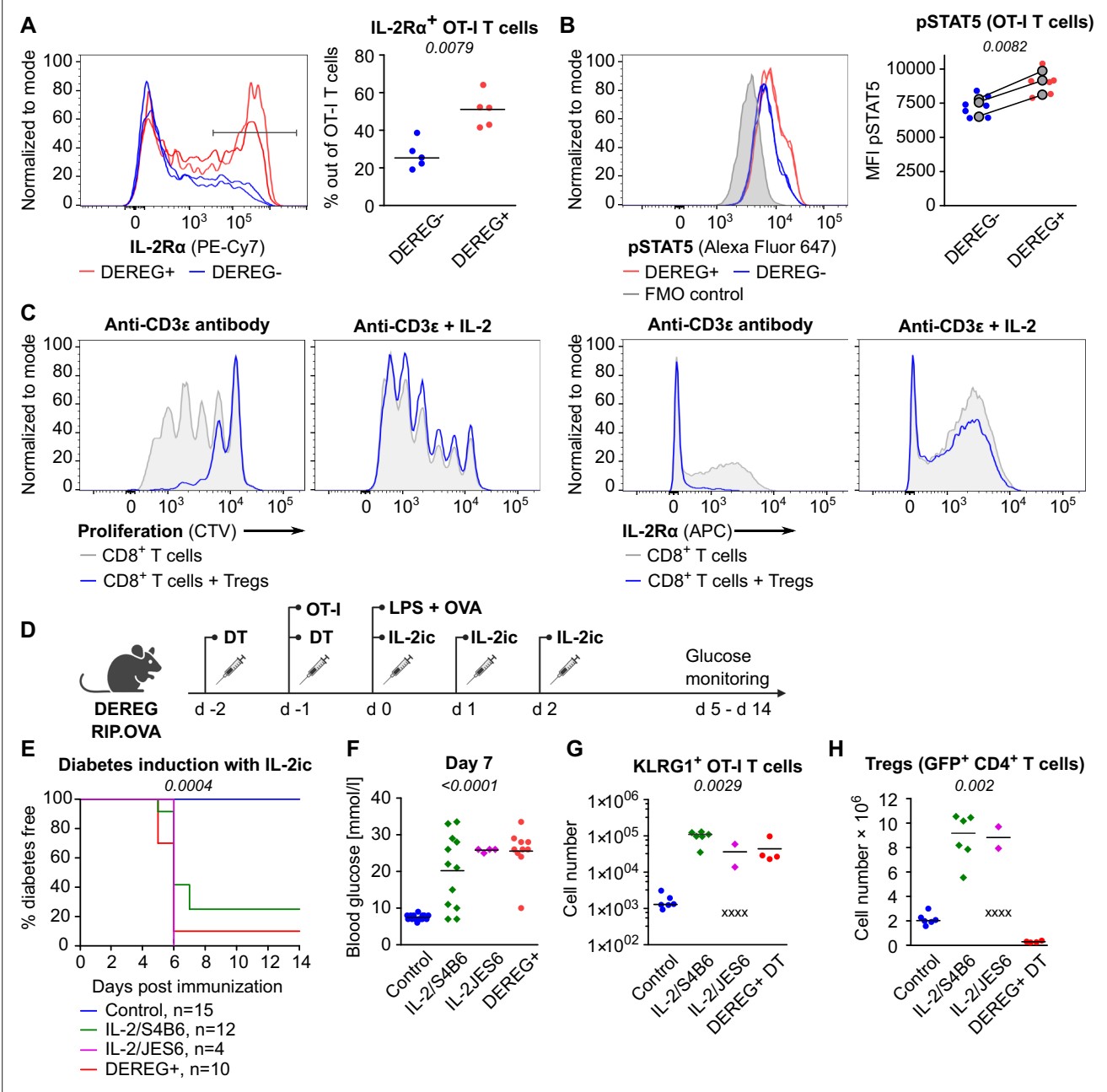

**Figure 4.** Tregs maintain tolerance of CD8+ T cells via limiting IL-2. (**A, B**) OT-I T cells (5 × 10⁴) were transferred into Treg-depleted DEREG+ RIP.OVA mice and control DEREG- RIP.OVA mice. The next day, mice were immunized with DC-OVA. On day 3 post-immunization, spleens were collected and analyzed by flow cytometry. OT-I T cells were identified as OVA-tetramer and TCRVα2 double positive. (**A**) IL-2Rα expression on OT-I T cells. Left: a representative experiment out of three in total. Right: n = 5 mice per group. Median is shown. (**B**) pSTAT5 expression in OT-I T cells. Left: a representative experiment out of three in total. Right: geometric mean fluorescence intensity (MFI) of anti-pSTAT5-Alexa Fluor 647 in OT-I T cells. Mean of MFI values for each group per experiment are shown as a gray dots. Lines connect data from corresponding experiments. DEREG- n = 7, DEREG+ n = 6. (**C**) In vitro proliferation assay. CTV-labeled CD8+ T lymphocytes were stimulated with plate-bounded anti-CD3ε-antibody in the presence or absence of Tregs and/or recombinant IL-2. After 72 hr, cells were analyzed by flow cytometry. Left: proliferation is indicated by the CTV dilution. Right: IL-2Rα expression on CD8+ T cells. A representative experiment out of four in total. (**D–F**) Treg-depleted DEREG+ RIP.OVA mice and control DEREG- RIP.OVA mice received OT-I T cells, followed by the immunization with OVA peptide and LPS. On days 0, 1, and 2 post-immunization, RIP.OVA mice received IL-2ic (IL-2/S4B6 or IL-2/JES6). (**D**) Experimental scheme. (**E**) Urine glucose level was monitored on a daily basis for 14 days. Percentage of diabetes-free mice is shown. Number of mice is indicated. Four mice from IL-2/JES6 group died before day 5 and were excluded from the analysis. (**F**) Blood glucose concentration on day 7 post-immunization. Number of mice is indicated in (**E**). (**G, H**) Diabetes was induced in Treg-depleted DEREG+ RIP.OVA mice (with DT) or Treg-replete DEREG+ RIP.OVA mice (without DT), which later received IL-2ic (IL-2/S4B6 or IL-2/JES6), or were left untreated (control) (scheme

*Figure 4 continued on next page*

*Figure 4 continued*

D). On day 5 post-immunization, spleens were collected and analyzed by flow cytometry. Four mice from IL-2/JES6 group died before the analysis (shown as 'x'). Control n = 6, IL-2/S4B6 n = 6, IL-2/JES6 n = 2, DEREG+ DT n=4. (**G**) Number of KLRG1+ OT-I T cells. Median is shown. (**H**) Number of Tregs (defined as GFP+ CD4+ T cells). Median is shown. Statistical significance was calculated by two-tailed Mann–Whitney test (comparison of two groups), Kruskal–Wallis test (comparison of four groups), or log-rank (Mantel–Cox) test (survival), p-value is shown in italics.

The online version of this article includes the following figure supplement(s) for figure 4:

**Figure supplement 1.** The role of IL-2 in the suppression of CD8+ T cells by Tregs.

IL-2ic provided during OT-I priming (*Figure 4D*). IL-2ic have markedly increased biological activity in comparison to IL-2 alone (*Boyman et al., 2006*). We used IL-2ic with JES6 antibody clone, which selectively promotes IL-2 signaling in T cells expressing high-affinity IL-2Rαβγ trimeric IL-2 receptor, or with the antibody clone S4B6, which promotes IL-2 signaling irrespective of IL-2Rα expression (*Boyman et al., 2006*; *Spangler et al., 2015*). Both clones of IL-2ic broke the peripheral tolerance and induced the onset of autoimmune diabetes similarly to the Treg-depletion (*Figure 4E and F*). IL-2ic promoted the formation of KLRG1+ effector OT-I T cells (*Figure 4G*), despite a dramatic IL-2-dependent expansion of Tregs at the same time (*Figure 4H*). Overall, these results showed that Tregs are unable to suppress CD8+ T cells in the excess of IL-2 signal, although Tregs themselves expand dramatically, which should augment their overall suppressive capacity with the exception of the limiting IL-2 mechanism. This supports the hypothesis that the major mechanism of Treg-mediated suppression of self-reactive CD8+ T cells is the reduction of IL-2 availability. This IL-2-dependent mechanism might be specific for the suppression of CD8+ T cells since antigen-activated CD8+ T cells were much more responsive to IL-2 than activated CD4+ T cells (*Figure 4—figure supplement 1F*).

## Tregs suppress formation of a unique subset of CD8+ effector T cells

To study how Treg-mediated suppression alters the gene expression profiles of activated CD8+ T cells, we analyzed the transcriptomes of OT-I T cells primed in the presence or absence of Tregs on a single-cell level. IL-2-responsive genes were upregulated in OT-I T cells primed in the Treg-depleted mice, including those encoding the key cytotoxic molecules granzyme B and perforin, as well as IL-2Rα (*Figure 5A*, *Figure 5—figure supplement 1A and B*). Unsupervised clustering highlighted five distinct activated OT-I T cell subsets (*Figure 5B*). Based on their gene expression profiles (*Figure 5—figure supplement 1C*, *Supplementary file 1*) and comparison with CD8+ T-cell subsets formed during an antiviral response (*Figure 5—figure supplement 1D*), we identified clusters 0 and 1 as early effector T cells, cluster 2 as memory T cells precursors, and cluster 3 as precursors of exhausted T cells. Cluster 4, on the other hand, did not match to any established CD8+ T-cell subset. It was characterized by the expression of a natural killer (NK) cell activation receptor gene *Klrk1* (alias *Nkg2d*), *Gzmb* (*Figure 5C*, *Figure 5—figure supplement 1C*) and by the overall similarity to the NK cell gene expression profile (*Figure 5—figure supplement 1E*). Interestingly, these cells occurred almost exclusively in the absence of Tregs (*Figure 5D and E*). This cluster could be further divided into two subsets. The larger of them expressed *Il7r*, *Cd103*, and NK cell markers *Ifitm1-3* and *Cd7*, but not a typical effector T-cell marker *Cd49d (Itga4)*, which encodes a subunit of the integrin VLA4 (*Figure 5F–H*, *Figure 5—figure supplement 1C*). We refer to these cells as KLRK1+ IL-7R+ (KILR) effector CD8+ T cells in this study, reflecting their co-expression of typical cytotoxic effector genes (*Gzmb*, *Prf1*) together with markers of NK cells (*Klrk1*) or T-cell memory (*Il7r*) genes. In the next step, we validated the scRNAseq results using flow cytometry. This confirmed that KLRK1+ IL-7R+ CD49d- CD8+ T cells are ~20-fold more frequent in the absence of Tregs during the priming than in Treg-replete conditions (*Figure 5I*). This experiment also confirmed that part of these KLRK1+ IL-7R+ CD49d- cells expressed CD103 (*Figure 5—figure supplement 1F*). Moreover, the gene expression profile of FACS-sorted populations of OT-I T cells primed in the presence or absence of Tregs was consistent with the original scRNAseq data (*Figure 5—figure supplement 1G*).

## KILR effector CD8+ T cells have strong cytotoxic properties

In the next step, we tested whether Treg depletion induces KILR effector CD8+ T cells in polyclonal mice. Although Treg deficiency increased the frequency of KILR effector CD8+ T cells, they were present at low numbers (*Figure 6A*, *Figure 6—figure supplement 1A and B*). We hypothesized that only the combination of antigenic and IL-2 signals can effectively induce KILR effector CD8+ T cells. To

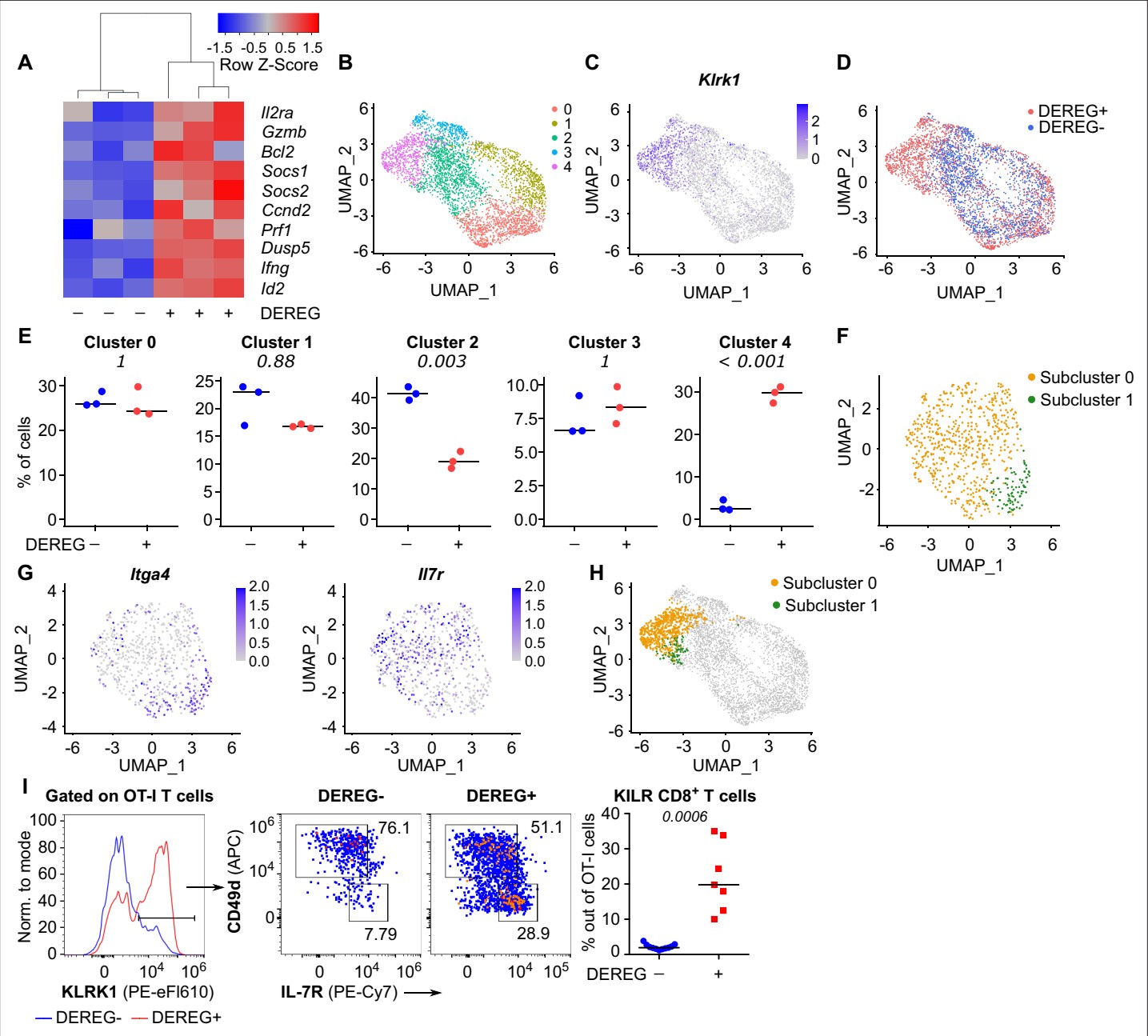

**Figure 5.** Tregs block the formation of KLRK1⁺ IL-7R⁺ cytotoxic T cells. (**A–E**) Ly5.1 OT-I T cells were adoptively transferred into Treg-depleted DEREG⁺ RIP.OVA and control DEREG⁻ RIP.OVA mice (n = 3 mice per group). The next day, mice were immunized with DC-OVA. On day 3 post-immunization, spleens were isolated and OT-I T cells were FACS sorted as Ly5.1⁺ CD8⁺ cells, and analyzed via scRNAseq. (**A**) A heat map showing the relative expression of canonical IL-2-responsive genes. Each column represents one mouse. (**B–D**) UMAP projection of the individual OT-I T cells based on their gene expression profile. (**B**) The colors indicate individual clusters revealed by unsupervised clustering. (**C**) The intensity of the blue color indicates the level of *Klrk1* expression in individual cells. (**D**) The origin of the cells (DEREG⁺ or DEREG⁻ mice) is indicated. (**E**) The percentage of cells assigned to specific clusters is shown for individual mice. Statistical significance was calculated by unpaired Student's *t*-test with Bonferroni correction, p-value is shown in italics. Median is shown. (**F–H**) Cluster 4 was reanalyzed separately. (**F**) UMAP projection showing two subclusters identified by unsupervised clustering. (**G**) The intensity of the blue color indicates the level of expression of *Itga4* or *Il7r* in individual cells. (**H**) Projection of subclusters 0 and 1 on the original UMAP plot. (**I**) Ly5.1 OT-I T cells (5× 10⁴) were adoptively transferred into Treg-depleted DEREG⁺ RIP.OVA mice (n = 7) or DEREG⁻ RIP.OVA mice (n = 11). The next day, mice were immunized with DC-OVA. On day 3 post-immunization, spleens were collected and analyzed by flow cytometry. Left: a representative experiment out of three in total is shown. KLRK1⁺ subset of Ly5.1 OT-I T cells was divided into two gates based on their expression of CD49d and IL-7R. Percentage of KILR effector T cells, defined as KLRK1⁺ IL-7R⁺ CD49d⁻ cells out of OT-I T cells is shown. Statistical significance was calculated by two-tailed Mann–Whitney test, p-value shown in italics. Median is shown.

*Figure 5 continued on next page*

*Figure 5 continued*

The online version of this article includes the following figure supplement(s) for figure 5:

**Figure supplement 1.** The comparison of the differentiation of CD8+ T cells primed in the presence and absence of Tregs in vivo.

address this hypothesis, we directly treated the OT-I *Rag2*−/− mice with OVA and/or IL-2ic. Indeed, the combination of IL-2 administration and antigenic stimulation efficiently induced KILR effector CD8+ T cells in OT-I mice (*Figure 6B–E*), which were characterized by high GZMB and IL-7R expression, whereas the stimulation with IL-2 or antigen alone failed to induce the complete KILR phenotype. The combination of IL-2ic and OVA also induced the expression of IL-2Rα (*Figure 6—figure supplement 1C*).

We addressed the stability of the KILR phenotype by co-transferring congenically marked KILR (CD49d− KLRK1+) and naïve (CD49d− KLRK1−), or KILR and effector (CD49d+ KLRK1−) OT-I T cells at 1:1 ratio into T-cell-deficient *Cd3e*−/− mice and analyzed the counts and phenotypes of these cells after 3 days (*Figure 6—figure supplement 1D*). We observed that all three subsets maintained the expression of their signature markers (*Figure 6F*), documenting the stability of the KILR phenotype. KILR cells slightly outcompeted effector T cells (*Figure 6G*, *Figure 6—figure supplement 1E*). This can be explained by higher frequency of apoptotic cells together with slightly reduced proliferation in effector than in KILR cells (*Figure 6—figure supplement 1F and G*).

To assess the cytotoxic properties of these cells, we compared the ability of KLRK1+ and conventional KLRK1− effector T cells from OT-I mice treated with OVA + IL-2ic, and KLRK1+ T cells from mice treated only with OVA to kill splenocytes loaded with their cognate antigen in vivo (*Figure 6—figure supplement 1H*). KLRK1+ T cells induced by the combination of the antigen and IL-2ic showed the most potent cytotoxic activity on per cell basis (*Figure 6H*, *Figure 6—figure supplement 1I and J*).

To identify putative human counterparts of KILR effector CD8+ T cells, we generated a human blood CD8+ T-cell atlas by integrating publicly available single cell transcriptomic datasets from healthy donors, because CD8+ T cell data from Treg-deficient patients are not available. After removing MAIT cells (*Figure 6—figure supplement 1K*, cluster 6), we identified naïve, memory, and effector T cells, and the population expressing some KILR effector CD8+ T-cell signature genes such as NK markers (*KLRD1, IFITM3, CD7*) and *IL7R* (*Figure 6I and J*, *Figure 6—figure supplement 1L*, *Supplementary file 2*). Similarly to mouse KILR effector CD8+ T cells, this human T-cell subset was enriched for NK signature genes (*Figure 6—figure supplement 1M*). Human KILR-like T cells constituted for ~1–10% of all CD8+ T cells (*Figure 6K*). These cells showed no signs of clonal expansion, but expressed rearranged αβTCR genes, ruling out the possibility that these cells were NK cells or another non-T cell subset (*Figure 6L*). Because these cells expressed high levels of *IL2RB* (*Figure 6J*), a subunit of IL-2 and IL-15 receptors, their gene expression profile was probably modulated by strong IL-2/IL-15 signals, which is in line with the origin of mouse KILR effector T cells. Although human KILR-like T cells expressed cytotoxic genes *GNLY* and *GZMK*, the expression of *GZMA* and *GZMB* was lower in these cells than in the conventional effector cells (*Figure 6J*). This probably reflects the fact that these T cells have not been stimulated by IL-2 together with their cognate antigen recently, which is required for the formation of the full KILR phenotype in mice (*Figure 6B–E and H*). Nevertheless, this previously uncharacterized population of human CD8+ T cells shows apparent similarities to mouse KILR effector CD8+ T cells induced by supra-physiological IL-2 levels.

## Strong IL-2 signal promotes anti-tumor CD8+ T-cell responses

So far, we have documented that Tregs suppress self-reactive CD8+ T cells by limiting IL-2 signal in the context of the autoimmune pathology. We hypothesized that excessive IL-2 signaling may override Treg-mediated suppression of tumor-reactive CD8+ T cells as well. Therefore, we investigated the effect of IL-2/JES6 immunocomplexes which were previously considered tolerogenic, in a BCL1 leukemia model (*Figure 7A*). This IL-2ic slowed down the progression of the disease, when co-administrated with a chemotherapeutic drug doxorubicin (Dox) (*Figure 7B*, *Figure 7—figure supplement 1A*). This effect was dependent on CD8+ but not on CD4+ T cells, as revealed by antibody-mediated depletion of these subsets (*Figure 7B*). Depletion of CD4+ T cells even more improved the anti-tumor effect of Dox plus IL-2ic combinational treatment, presumably due to the depletion of Tregs, which could still limit IL-2 availability between the IL-2ic injections. The administration of IL-2ic without the chemotherapy showed no therapeutic effect (*Figure 7B*, *Figure 7—figure supplement 1A*),

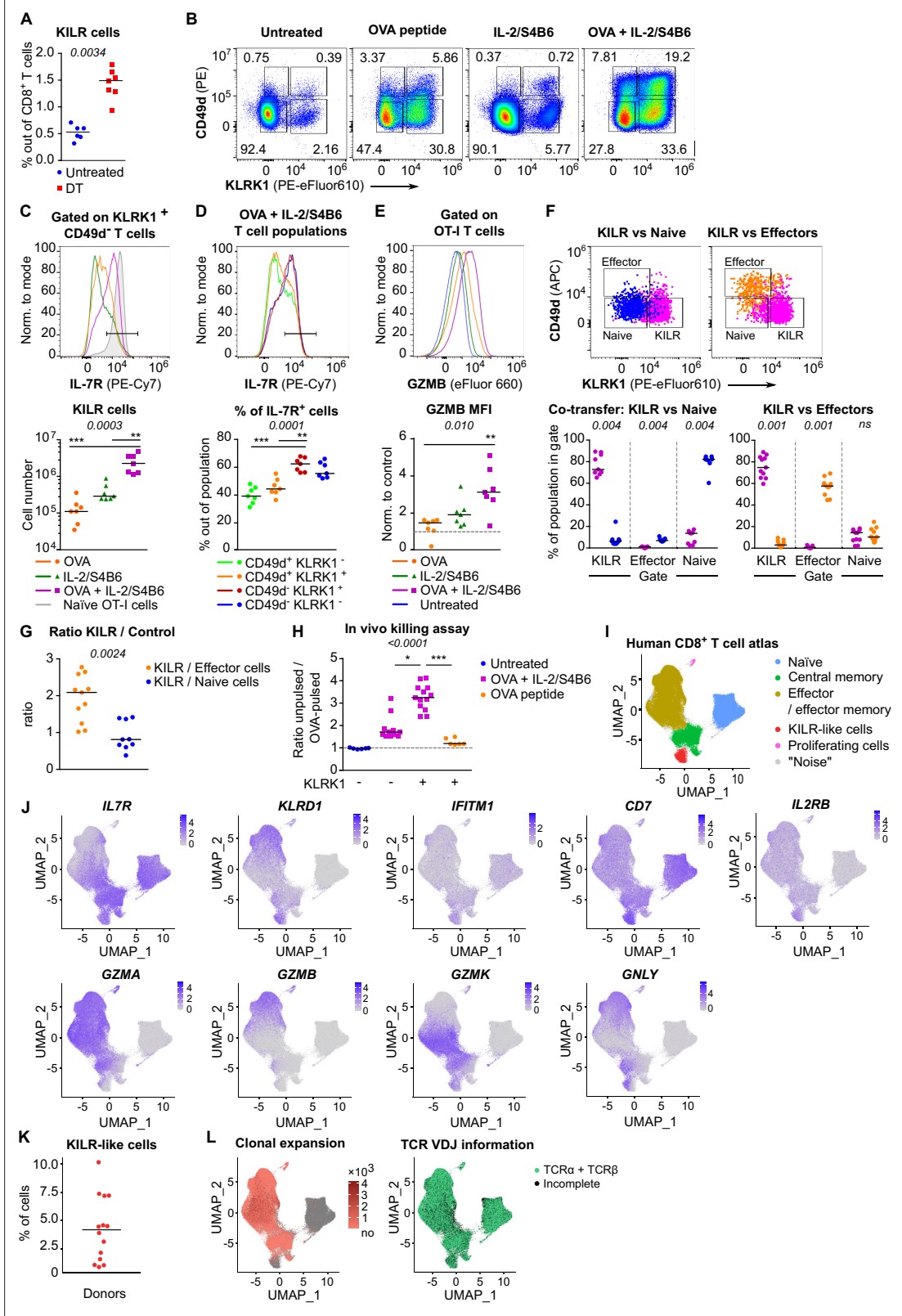

**Figure 6.** KILR CD8+ T cells induced by the stimulation with the cognate antigen and high levels of IL-2 show superior cytotoxicity. (**A**) DEREG+ RIP.OVA mice were treated with DT in order to deplete Tregs on days 0 and 1. On day 3, spleens were collected and analyzed by flow cytometry. Percentage of KILR effector T cells defined as KLRK1+ CD49d- IL-7R+ among CD8+ T cells is shown. Untreated n = 6, DT n = 7. Median is shown. (**B–E**) OT-I *Rag2-/-* mice were treated with OVA peptide (single dose on day 0, n = 7) and/or IL-2/S4B6 (three doses, days 0, 1 and 2, n = 7 mice per group) or left untreated (n

*Figure 6 continued on next page*

*Figure 6 continued*

= 6). Spleens were collected and analyzed by flow cytometry on day 3. (**B**) Four populations of OT-I T cells were identified based on their expression of KLRK1 and CD49d. A representative experiment out of three in total. (**C**) Top: IL-7R expression on KLRK1+ CD49d- OT-I T cells from mice treated with OVA and/or IL-2/S4B6. Expression of IL-7R on naïve OT-I T cells is shown as a positive control. Representative staining. Bottom: number of KILR effector T cells, defined as KLRK1+ CD49d- IL-7R+ OT-I T cells. Median is shown. (**D**) IL-7R expression in CD49d+ KLRK1-, CD49d+ KLRK1+, CD49d- KLRK1+, and CD49d- KLRK1- OT-I T cells from mice treated with OVA + IL-2/S4B6. Top: a representative histogram. Bottom: percentage of IL-7R+ cells among OT-I T cells in indicated populations. Median is shown. (**E**) GZMB levels in OT-I T cells. Top: a representative histogram. Bottom: geometric mean fluorescence intensity (MFI) of anti-GZMB-eFluor 660 antibody on OT-I T cells. Obtained values were normalized to the average of MFI of untreated samples in each experiment (=1). Median is shown. (**F–G**) OT-I *Rag2-/-* and Ly5.1 OT-I *Rag2-/-* mice were treated with OVA peptide (day 0) and IL-2/S4B6 (three doses, days 0–2). On day 3, KILR (KLRK1+ CD49d-), naïve (KLRK1- CD49d-), and effector cells (KLRK1- CD49d+) were sorted. Recipient *Cd3e-/-* mice received a mix of KILR and naïve (n = 9), or KILR and effector (n = 11) congenic cells (1:1 ratio, 400 × 10³ or 500 × 10³ cells in total). On day 7, spleens of the recipient mice were analyzed by flow cytometry. Two independent experiments were performed. (**F**) Percentage of cells that kept their initial phenotype was determined. Top: cells sorted as KILR (magenta), naïve (blue), and effector cells (orange) fall into corresponding gates. Representative dot plot. Bottom: percentage of cells that fall into the KILR, effector, and naïve gate, after adoptive co-transfer. Median is shown. (**G**) Ratio of KILR cells to co-transferred control cells. Median is shown. (**H**) OT-I *Rag2-/-* mice were treated with OVA peptide (day 0), and/or IL-2/S4B6 (days 0, 1, and 2) or left untreated. Spleens were collected on day 3. KLRK1+ CD8+ or KLRK1- CD8+ cells were sorted and adoptively transferred into recipient RIP.OVA mice, which have received a mixture of target OVA-pulsed CTV-loaded and unpulsed CFSE-loaded splenocytes from Ly5.1 mice at ~1:1 ratio earlier on the same day. The next day, the spleens were analyzed for the presence of Ly5.1 donor cells by flow cytometry. Ratio of unpulsed (CFSE+) to OVA pulsed (CTV+) target cells was determined and normalized to control recipients which did not receive OT-I T cells (=1). KLRK1- (Untreated) n = 6, KLRK1- (OVA + IL-2/JES6) n=11, KLRK1+ (OVA + IL-2/JES6) n = 13, KLRK1+ (OVA peptide) n = 6. Four independent experiments. Median is shown. (**I–L**) Human CD8+ T-cell atlas was generated by integrating 14 scRNAseq data sets from blood of healthy donors. The gene expression data after the removal of MAIT cells were projected into a 2D UMAP plot. (**I**) The assignment of individual cells to clusters identified by unsupervised clustering. Individual clusters were matched to established CD8+ T cell subsets based on the expression of their signature markers (see *Figure 6—figure supplement 1H*). (**J**) The intensity of the blue color corresponds to the level of expression of indicated genes in individual cells. (**K**) The percentage of CD8+ T cells assigned to the KILR-like CD8+ T-cell cluster in individual donors (n = 14). Median is shown. (**L**) Left: clonally expanded T cells were identified based on their TCRαβ VDJ sequences. The intensity of the red color indicates the size of individual clones. Right: T cells with recovered complete TCRαβ VDJ information are shown in green. Statistical significance was calculated by two-tailed Mann–Whitney test (**A, G**), Wilcoxon matched-pairs signed rank test (**F**), or Kruskal–Wallis test (**C, D, E, H**) for multiple groups comparison (p-value is shown in italics) with Dunn's post-test (*<0.05, **<0.01, ***<0.001).

The online version of this article includes the following figure supplement(s) for figure 6:

**Figure supplement 1.** The characterization of IL-2-induced KILR CD8+ T cells and human blood KILR-like T cells.

suggesting a synergy between the IL-2 signals and the presentation of antigens released from tumor cells undergoing an immunogenic cell death (*Obeid et al., 2007*).

We analyzed the effect of the combinatorial therapy on the formation of different effector subsets. Mice treated with Dox and IL-2ic, but not those treated with Dox alone or untreated, generated a large population of KLRK1+ T cells (*Figure 7C*). The frequencies of KLRK1+ CD49d+, KLRK1+ CD49d-, KRLK1+ IL-7R+, and KLRK1+ GZMB+ CD8+ T-cell subsets were significantly higher in the spleens of mice treated with Dox and IL-2ic than in the control mice (*Figure 7C*, *Figure 7—figure supplement 1B*). This analysis showed that the formation of KILR T cells and KLRK1+ T-cell in general correlates with the improved survival of leukemic mice.

We observed very similar protective effects of combinational therapy of Dox plus IL-2/JES6 in a B16F10 melanoma model (*Figure 7D and E*, *Figure 7—figure supplement 1C and D*). The spleens of mice treated with Dox and IL-2ic had much higher frequencies of KLRK1+ GZMB+, KLRK1+ CD49d+, and KLRK1+ CD49d- CD8+ T cells than the untreated or Dox-only treated mice (*Figure 7F*, *Figure 7—figure supplement 1E and F*). A large proportion of CD8+ T cells infiltrating the tumors in the Dox plus IL-2ic treated mice had the KLRK1+ CD49d- phenotype, whereas these cells were relatively rare in the untreated and Dox-only treated mice. Moreover, KLRK1+ T cells in tumors of Dox + IL-2ic group expressed higher levels of IL-7R and GZMB than their counterparts in untreated or Dox-only treated mice (*Figure 7H*).

Overall, the Dox + IL-2ic therapy enhanced the anti-tumor immune response in a CD8+ T cell-dependent manner and induced large numbers of KLRK1+ CD8+ T cells. A substantial part of these KLRK1+ CD8+ T cells, especially in the tumor, were *bona fide* KILR cells as shown by their high expression of IL-7R and GZMB and low expression of CD49d.

Since IL-2/JES6 highly selectively stimulates IL-2Rα+ cells, represented mostly by Tregs in naïve unprimed mice, it was considered as an immunotherapeutic tool for the specific expansion of Tregs in vivo with a potential application in the treatment of autoimmune diseases and pre-transplantation care. On the contrary, our data shows that it promotes antitumor response of CD8+ T cells, particularly

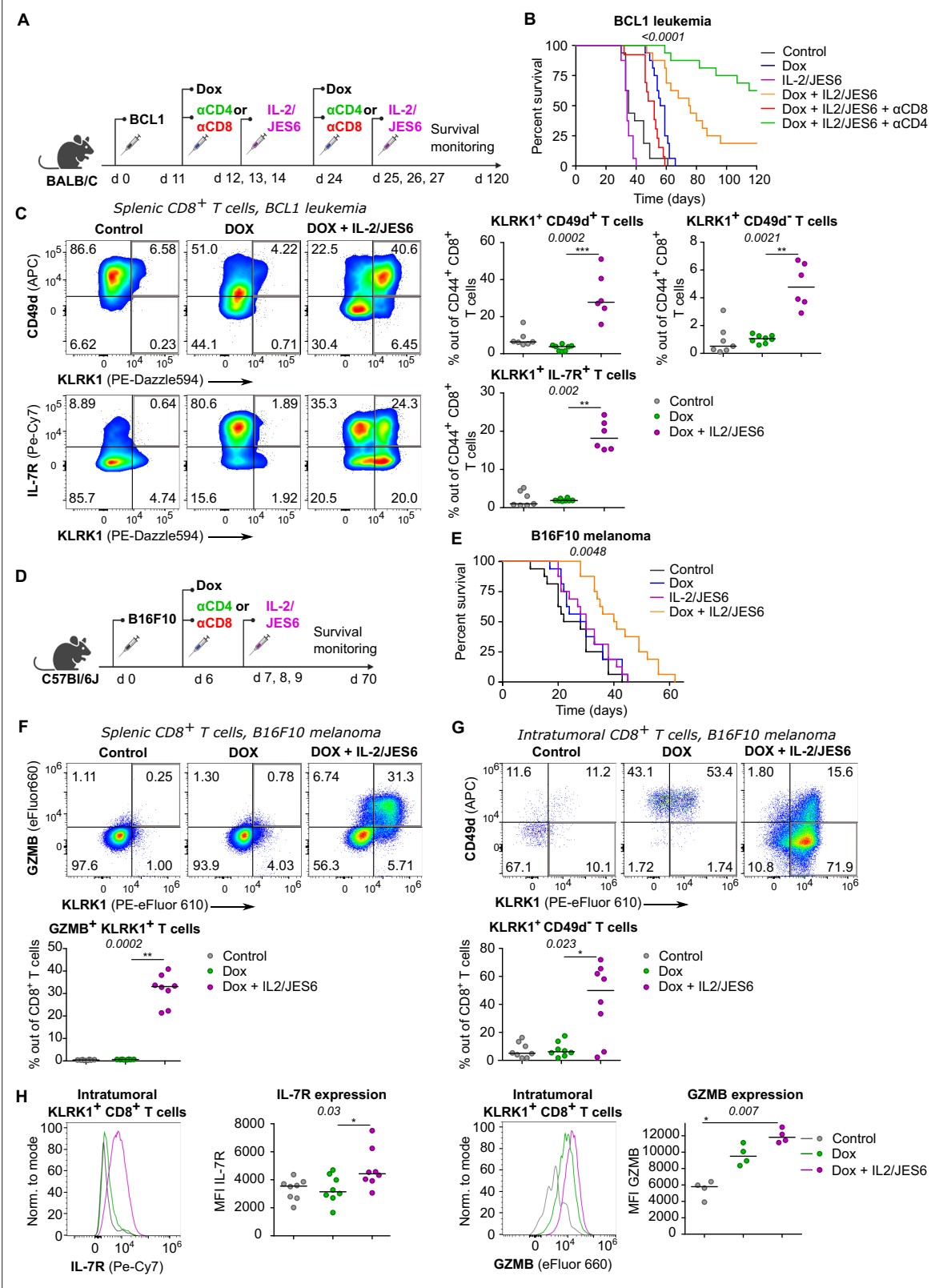

**Figure 7.** A combined treatment of IL-2/JES6 immunocomplexes and chemotherapy hampers the tumor growth and induces KILR CD8+ T cells. (**A–C**) On day 0, BALB/C mice were inoculated with BCL1 leukemia cells. On days 11 and 24 post inoculation, mice received doxorubicin (Dox), with or without anti-CD4 or anti-CD8 depletion mAbs. On three consecutive days, mice were treated with IL-2/JES6. (**A**) Scheme of the experiment. (**B**) Survival curves. n = 16 mice per group in two independent experiments with the exception of "Dox + IL-2/JES6 + αCD8" conditions where three mice in the second

*Figure 7 continued on next page*

**Figure 7 continued**

experiment did not tolerate anti-CD8 administration for unknown reasons and were removed. (**C**) On day 30, spleens of mice from control (n = 7), Dox-treated (n = 8), and Dox + IL-2/JES6-treated (n = 6) groups were collected and analyzed by flow cytometry. Left: expression of KLRK1, CD49d, and IL-7R on CD44$^+$ CD8$^+$ T cells. Representative staining. Right: percentage of KLRK1$^+$ CD49d$^-$, KLRK1$^+$ CD49d$^+$, and KLRK1$^+$ IL-7R$^+$ cells out of CD44$^+$ CD8$^+$ T cells. Two independent experiments. (**D–H**) On day 0, C57Bl/6J mice were inoculated with B16F10 melanoma cells. On day 6, mice received doxorubicin (Dox). On three consecutive days, mice were treated with IL-2/JES6. (**D**) Scheme of the experiment showing also the eventual treatment with anti-CD4 or anti-CD8 depletion mAb relevant for *Figure 7—figure supplement 1B*. (**E**) Survival curves. n = 16 mice per group in two independent experiments. (**F–H**) On day 11, spleens and tumors of mice from control, Dox-treated, and Dox + IL-2/JES6-treated groups (n = 8 per group) were collected and analyzed by flow cytometry. Two independent experiments. (**F**) Top: expression of KLRK1 and granzyme B on splenic CD8$^+$ T cells. Representative dot plot. Bottom: percentage of KLRK1$^+$ GZMB B$^+$ cells out of total CD8$^+$ T cells. (**G**) Top: expression of KLRK1 and CD49d on intratumoral CD8$^+$ T cells. Representative dot plot. Bottom: percentage of KLRK1$^+$ CD49d$^-$ cells out of total CD8$^+$ T cells. (**H**) IL-7R and GZMB expression on intratumoral KLRK1$^+$ CD8$^+$ T cells. Representative histograms and geometric mean fluorescence intensity (MFI) are shown. Statistical significance was calculated by log-rank (Mantel–Cox) test (survival **B**, **E**), or Kruskal–Wallis test (**C, F–H**) (p-value is shown in italics) with Dunn's post-test (*<0.05, **<0.01, ***<0.001). Median is shown.

The online version of this article includes the following figure supplement(s) for figure 7:

**Figure supplement 1.** The effects of combined treatment of tumor-bearing mice with IL-2/JES6 immunocomplexes and doxorubicine.

in the combination with the immunogenic chemotherapy (*Obeid et al., 2007*; *Kim and Kin, 2021*), and that this anti-tumor activity is not counteracted by Tregs.

## Discussion

In this study, we investigated how Tregs prevent CD8$^+$ T-cell mediated autoimmune pathology. Using a well-controlled system based on the transfer of CD8$^+$ T cells specific for a pancreatic neo-self-antigen, we revealed that regulatory T cells substantially increase the quorum of self-reactive T cells (*Bosch et al., 2017*) required to induce the autoimmune pathology.

We observed that Tregs do not alter the antigen-affinity discrimination as they suppress both high-affinity and low-affinity CD8$^+$ T cells, which is in an apparent contrast with a previous study (*Pace et al., 2012*) concluding that Tregs suppress exclusively low-affinity T-cell responses. The most possible explanation is that the previous study focused on the role of Tregs in the CD8$^+$ T-cell response to *Listeria monocytogenes*, where the presence or depletion of Tregs might regulate the kinetics of pathogen clearance and thus the dose of bacterial antigens. Contrary to that, the dose of the priming antigen was constant in our experimental diabetes.

We have identified the reduction of available IL-2 as the major mechanism of Treg-mediated tolerance of CD8$^+$ T cells. The evidence is that Treg depletion increases IL-2 signaling in these cells, i.e., pSTAT5 levels and expression of IL-2-responsive genes, and that the administration of IL-2 and IL-2ic mimics the absence of Tregs in vitro and in vivo, respectively. Strong exogenous IL-2 signals stimulate the expansion of Tregs and the expression of their effector molecules (*Tomala and Kovar, 2016*), which should enhance all their potential suppression mechanisms with the notable exception of IL-2 sequestration. However, as this exogenous IL-2 signal mimics the effect of Treg depletion in our model, mechanisms non-targeting IL-2 do not seem to contribute substantially to the suppression of self-reactive CD8$^+$ T cells. This conclusion is in line with some previous studies of Treg-mediated suppression of effector CD8$^+$ T cells (*Kalia et al., 2015*; *McNally et al., 2011*) and with the importance of IL-2Rα expression in self-reactive CD8$^+$ T cells for the infiltration of pancreatic islets (*Marchingo et al., 2014*).

It has been found that the biological activity of IL-2ic is dictated by the anti-IL-2 mAb clone (*Boyman et al., 2006*; *Spangler et al., 2015*). IL-2/JES6 immunocomplexes selectively stimulate cells expressing the high-affinity trimeric IL-2 receptor and were considered as an immunosuppressive agent acting via the expansion of Tregs (*Tomala and Kovar, 2016*). In contrast, our conclusion that IL-2 restriction is the major mechanism of Treg-mediated suppression of CD8$^+$ T cells implies that the administration of IL-2/JES6 should release antigen-activated CD8$^+$ T cells from the Treg control. Indeed, IL-2/JES6 in combination with chemotherapy significantly prolonged the survival of tumor-bearing mice in two different tumor models in a CD8$^+$ T-cell-dependent manner. Accordingly, the administration of IL-2/JES6 after priming, but not before the transfer of self-reactive T cells, decreased the T-cell quorum for the induction of experimental diabetes. Collectively, these results reveal that a strong sustained IL-2 signal, even if selective for IL-2Rα$^+$ cells, potentiates antigen-induced CD8$^+$ T-cell antitumor immunity

and autoimmunity despite its concomitant stimulatory effects on Tregs. Recently, a combinational therapy of in vitro expanded polyclonal Tregs and low-dose IL-2 was tested in patients with type I diabetes in a phase I clinical trial (**Dong et al., 2021**). The authors of the study did not observe a benefit of this intervention for the patients but they reported an expansion of Tregs along with activated NK and cytotoxic CD8+ T cells, which corresponds to our conclusions.

Our finding that IL-2 restriction is the major mechanism of Treg-mediated regulation of self- and tumor-reactive CD8+ T cells does not exclude the involvement of additional mechanisms regulating other immune cell types, such as conventional CD4+ T cells. On the contrary, there is substantial evidence that Tregs suppress conventional CD4+ T cells largely via mechanisms not dependent on IL-2 sequestration. First, we show that antigen-stimulated CD4+ T cells are less sensitive to IL-2 signals than CD8+ T cells in vivo, which limits the potential impact of the IL-2 restriction on them. Second, it has been shown that Tregs lacking the high-affinity IL-2 receptor, implied in IL-2 sequestration, can still control the homeostasis of CD4+ T cells but not CD8+ T cells (**Chinen et al., 2016**). Third, it has been shown that Tregs suppress conventional CD4+ T cells by depleting their cognate peptide-MHCII from APC, which is not applicable for the suppression of MHCI-restricted CD8+ T cells (**Akkaya et al., 2019**). In contrast, a recent study proposed that self-reactive CD4+ T cells produce IL-2 locally attracting Tregs to their close proximity, which leads to the suppression of later phases of self-reactive T-cell activation at least partially via limiting IL-2 (**Wong et al., 2021**).

Antigenic stimulation of CD8+ T cells in the presence of excessive IL-2, induced by depletion of Tregs or by administration of exogenous IL-2R agonists, leads to the formation of a previously uncharacterized subset of KILR effector CD8+ T cells. They express high levels of cytotoxic molecules, such as granzymes, but unlike conventional effector CD8+ T cells, they also express high levels of IL-7R. Moreover, they express several NK receptors, including KLRK1/NKG2D, which is involved in target cell killing (**Prajapati et al., 2018**). Since these cells show superior cytotoxic properties in vivo, suppression of their formation is probably a major mechanism of Treg-mediated self-tolerance. On the other hand, the existence of the KILR effector gene expression program suggests the possibility that these cells might arise naturally in specific immunological conditions, such as some type of infection. Indeed, it has been shown that the infection with lymphocytic choriomeningitis virus induces a transient drop in Treg numbers (**Schorer et al., 2020**), which might alleviate the Treg-mediated suppression of virus-specific CD8+ T cells in the early phase of the immune response. However, we could not find KILR effector CD8+ T cells in any scRNAseq dataset from the course of mouse infection, suggesting that such KILR effector-inducing conditions would be rather rare. On the other hand, we found increased frequencies of KILR cells in the spleens and tumors in mice with cancer treated with Dox and IL-2ic. The antigenic stimulation was probably provided by tumor-associated antigens released from cancer cells undergoing the immunogenic cell death induced by Dox (**Obeid et al., 2007**; **Kim and Kin, 2021**). The expression of IL-7Rα in KILR effector CD8+ T cells is striking as conventional effector T cells are characterized as IL-7R-, which is associated with their short lifespan (**Joshi et al., 2007**; **Kaech et al., 2003**). A previous study showed that antigenic stimulation together with activating antibodies to OX40 and 4-1BB TNF-family receptors induced IL-7R+ effector CD8+ T cells (**Lee et al., 2007**). These cells produced high levels of proinflammatory cytokines in an IL-7-dependent manner (**Lee et al., 2007**). Since 4-1BB signaling has been shown to enhance IL-2 production and IL-2Rα expression in T cells (**Barsoumian et al., 2016**; **Oh et al., 2015**), it is possible that those IL-7R+ effector cells were induced via a strong IL-2 signal. However, as the gene expression profile of OX40/4-1BB-induced IL-7R+ effector CD8+ T cells has not been analyzed (**Lee et al., 2007**), it is not clear whether or not these cells have other similarities to KILR effector CD8+ T cells.

Our reanalysis of publicly available datasets of CD8+ T cells revealed KILR-like CD8+ T cells in the human peripheral blood. These cells show patterns of KILR T-cell gene expression, but they do not express high levels of cytotoxic molecules, probably because they lack recent antigenic/IL-2 stimulation. Therefore, future studies are needed to resolve their phenotype and function. In any case, the potent cytotoxic capacity and their expansion by IL-2 agonists make KILR effector CD8+ T cells a promising clinical target in cancer immunotherapy.

## Materials and methods

### Antibodies, peptides, and dyes

Antibodies to the following antigens were purchased from BioLegend and used for flow cytometry: CD4 BV650 (RM4-5, #100545), CD4 Alexa Flour 700 (RM4-5, #100536), CD4 APC-Cy7 (GK15, #100414), CD4 APC (RM4-5, #100516), CD8a BV421 (53-6.7, #100738), CD8a PE (53-6.7, #100708), CD11c Alexa Flour 700 (N418, #117319), CD25 Alexa Fluor 700 (PC61, #102024), CD25 BV605 (PC61, #102036), CD25 BV650 (PC61, #102038), CD25 PE-Cy7 (PC61, #102016), CD44 BV 650 (IM7, #103049), CD45.1 Alexa Fluor 700 (A20, # 110723), CD45.1 BV650 (A20, #110735), CD45.1 FITC (A20, #110706), CD45.1 PerCP-Cy5.5 (A20, #110728), CD45.2 Alexa Fluor 700 (104, #109822), CD45.2 APC (104, #109814), CD49d APC (R1-2, #103622), CD49d PE-Cy7 (R1-2, #103618), CD62L FITC (MEL-14, #104406), CD80 PerCP-Cy5.5 (16-10A1, #104722), CD86 Alexa Fluor 700 (GL-1, #105024), CD103 PerCP-Cy5.5 (2 E7, #121416), CD127 PE-Cy7 (A7R34, #135013), TCRβ APC (H57-597, #109212), TCRβ PE (H57-597, #109208), and KLRG1 BV510 (2F1/KLRG1, #138421). Antibodies to TCR Vα2 FITC (B20.1, #553288), CD45.1 APC (A20, #558701), CD45.1 PE (A20, #553776), and CD49d PE (R1-2, #553157) were purchased from BD Pharmingen. Antibodies to Granzyme B eFluor660 (NGZB, #50-8898-82), FOXP3 PE-Cy7 (FJK-16s, #25-5773-82), FR4 PE/Dazzle594 (12A5, #125016), KLRK1 PE-eFluor610 (CX5, #61-5882-82), and MHC class II (I-A/I-E) FITC (M5/114.15.2, #11-5321-82) were purchased from eBioscience.

Goat anti-rabbit IgG (H+L) secondary antibody conjugated to Alexa Fluor 647 from Invitrogen (#21245) was used following rabbit anti-mouse phospho-Stat5 (Tyr694) (D47E7) from Cell Signaling (#4322S). Anti-mouse CD3ε antibody (clone 145-2C11, #100302, BioLegend) was used for plate coating.

Primary antibodies that were used for immunohistochemistry: CD8a (EPR21769, #217344, Abcam), CD4 Alexa Flour 647 (RM4-5, #100530), and insulin (polyclonal, # PA1-26938, Invitrogen). Secondary antibodies: goat anti-rabbit IgG (H+L) Highly Cross-Adsorbed Secondary Antibody, Alexa Fluor 555 (Invitrogen, #A32732), goat anti-guinea pig IgG (H+L) Highly Cross-Adsorbed Secondary Antibody (Invitrogen, #A-11073).

Anti-CD4 (GK1.5, #BE0003-1) and anti-CD8α (53–6.7, #BE0004-1) depletion antibodies were purchased from Bioxcell (USA).

Anti-mouse IL-2 mAb S4B6 (#BE0043-1) and anti-mouse IL-2 mAb JES6-1A12 (#BE0043) used for preparation of IL-2/S4B6 and IL-2/JES6 complexes, respectively, were purchased from Bioxcell (USA).

OVA (SIINFEKL), Q4R7 (SIIQFERL), and Q4H7 (SIIRFEHL) peptides were purchased from Eurogentec or Peptides&Elephants.

CFSE (#65-0850-84) and Cell Trace Violet (CTV) (#C34557), LIVE/DEAD near-IR (#L10119), and Hoechst 33258 (#H3569) dyes were purchased from Invitrogen. FITC Annexin V Apoptosis Detection Kit with 7-AAD (#640922) was purchased from BioLegend.

### IL-2/S4B6 and IL-2/JES6 complexes

IL-2/S4B6 and IL-2/JES6 immunocomplexes were described previously (*Tomala and Kovar, 2016*). Complexes were prepared by mixing recombinant mouse IL-2 (Cat# 212-12, 100 µg/ml; PeproTech) with anti-IL-2 mAb at a molar ratio of 2:1 in PBS. After 15 min incubation at room temperature, the complexes were diluted in PBS into the desired concentration, frozen at –20°C, and thawed shortly before application.

### Mice

All the mice had C57Bl/6J or BALB/C background. DEREG (RRID: MMRRC_032050-JAX) (*Lahl et al., 2007*), *Foxp3^DTR* (RRID: IMSR_JAX:016958) (*Kim et al., 2007*), RIP.OVA (RRID: MGI:3789286) (*Kurts et al., 1998*), OT-I *Rag2^-/-* (RRID:MGI:3783776, MGI:2174910) (*Palmer et al., 2016*; *Shinkai et al., 1992*), OT-II *Rag2^-/-* (RRID: MGI:3762632, MGI:2174910) (*Shinkai et al., 1992*; *Barnden et al., 1998*), Ly5.1 (RRID: IMSR_JAX:002014) (*Jang et al., 2018*) strains were described previously. Mice were bred in specific-pathogen-free facilities (Institute of Molecular Genetics of the Czech Academy of Sciences, Prague; Department of Biomedicine, University Hospital, Basel) or in a conventional facility (Institute of Microbiology of the Czech Academy of Sciences, Prague). Mice were kept in the animal facility with a 12 hr of light–dark cycle with food and water ad libitum. Animal protocols were performed in accordance with the laws of the Czech Republic and Cantonal and Federal laws of Switzerland, and

approved by the Czech Academy of Sciences (identification no. 11/2016, 81/2018, 15/2019) or the Cantonal Veterinary Office of Baselstadt, Switzerland, respectively.

Males and females were used for the experiments. At the start of the experiment, all mice were 6–11 weeks old. If possible, age- and sex-matched pairs of animals were used in the experimental groups. If possible, littermates were equally divided into the experimental groups. No randomization was performed when the experimental groups were based on the genotype of the mice; otherwise mice were assigned to experimental groups randomly (defined by their ID numbers) prior to the contact between the experimenter and the mice. The experiments were not blinded since no subjective scoring method was used. The estimation of the sample size was based on our previous experience. We typically aimed at having 12 animals per group in three independent experiments. The final number of mice per group depended on the number of available mice with required genotype, number of isolated T-cells for adoptive transfers, and unpredictable events (e.g., mouse death). In cancer experiments, we aimed at 16 mice per group in two independent experiments.

ARRIVE Essential10 guidelines (https://arriveguidelines.org/) were followed. Besides the reported exclusion criteria described below, some very rare experiments were excluded, when we realized an unintended severe technical error/deviation from the protocol (a general pre-established criterium).

## RT-qPCR

$2–10 \times 10^4$ OT-I T lymphocytes were FACS sorted as CD8$^+$ CD45.1$^+$ CD49d$^-$ KLRK1$^-$ cells (naïve), CD8$^+$ CD45.1$^+$ CD49d$^+$ KLRK1$^-$ cells (antigen experienced), CD8$^+$ CD45.1$^+$ CD49d$^+$ KLRK1$^+$ cells (double positive), or CD8$^+$ CD45.1$^+$ CD49d$^-$ KLRK1$^+$ cells (KILR). Total RNA was isolated by TRIzol LS (Invitrogen, #10296010) and in-column DNase digestion using RNA Clean & Concentrator Kit (Zymo Research), according to manufacturer's instructions. RNA was stored at –80°C or transcribed immediately using RevertAid reverse transcriptase (Thermo Fisher Scientific, #EP0442) with oligo(dT)18 primers according to the manufacturer's instructions. RT-qPCR was carried out using LightCycler 480 SYBR green I master chemistry and a LightCycler 480 machine (Roche). All samples were measured in triplicates. Median CT values were normalized to a reference gene, Glyceraldehyde-3-Phosphate Dehydrogenase (*Gapdh*). The sequences of used primers are:

*Gapdh*: F TGCACCACCAACTGCTTAGC, R GGCATGGACTGTGGTCATGAG *mCd7*: F TGGATGCC CAAGACGTACA, R TAAGATCCCTTCCAGGTGCC *mIfitm1*: F ATGCCTACTCCGTGAAGTCTAGG, R GACAACGATGACGACGATGGC *mGzma*: F AAAGGACTCCTGCAATGGGG, R ATCGGCGATCTC CACACTTC *mGzmb*: F GGGGCCCACAACATCAAAGA, R GGCCTTACTCTTCAGCTTTAGCA *mGzmk*: F AAGGATTCCTGCAAGGGTGA, R ATTCCAGGCTTTTTGGCGATG *mKlrd1*: F TCGGTGGAGACT GATGTCTG, R AACACAGCATTCAGAAACTTCC *mIl7r*: F AAAGCCAGAGCGCCTGGGTG, R CTGG GCAGGGCAGTTCAGGC *mIl2ra*: F AGAACACCACCGATTTCTGG, R GGCAGGAAGTCTCACTCTCG *mCxcr6*: F ACTGGGCTTCTCTTCTGATGC, R AAGCGTTTGTTCTCCTGGCT

## Enrichment of T lymphocytes

T lymphocytes were enriched by negative selection using the Dynabeads Biotin Binder kit (Invitrogen, #11047), and biotinylated anti-CD19 (clone 1D3, ATCC# HB305), anti-CD4 (clone YTS177), or anti-CD8 antibodies (clone 2.43, ATCC# TIB-210), depending on the experimental setup. Antibodies were produced and biotinylated in house using (+)-Biotin N-hydroxysuccinimide ester (Sigma-Aldrich) in bicarbonate buffer. The excessive biotin was separated from the antibody using Sephadex G-25 (Sigma-Aldrich). For RT-qPCR experiments, CD8$^+$ T cells were enriched using Dynabeads Untouched Mouse CD8 Cells Kit (Invitrogen, #11417D) according to the manufacturer's instructions.

## Flow cytometry and cell sorting

Live cells were stained with relevant antibodies on ice. LIVE/DEAD near-IR dye or Hoechst 33258 were used for discrimination of viable and dead cells.

For intracellular staining of FOXP3 and Granzyme B, cells were fixed and permeabilized using Foxp3/Transcription Factor Staining Buffer Set (#00-5523-00, Invitrogen) according to the manufacturer's instructions. Fixed cells were stained at room temperature for 1 hr.

For intracellular staining of pSTAT5, splenic cells were fixed using Fixation/Permeabilization buffer (Foxp3/Transcription Factor Staining Buffer Set, #00-5523-00, Invitrogen) immediately after isolation at room temperature for 15 min, washed twice with permeabilization buffer (Foxp3/Transcription

Factor Staining Buffer Set, #00-5523-00, Invitrogen), washed with PBS, and stained with anti-pSTAT5 antibody at room temperature overnight. The next day, cells were stained with secondary antibody at room temperature for 1 hr.

Flow cytometry was carried out using an LSRII (BD Biosciences) or an Aurora (Cytek). Data were analyzed using FlowJo software (BD Biosciences).

Cell sorting was performed on an Influx or an Aria machines (both BD Biosciences).

## Histological analysis of pancreas

Pancreases were harvested from mice on day 4 post-immunization, mounted in OCT embedding compound (Sakura Finetek Tissue-Tek, #4583), and frozen at –80°C. 10-μm-thick tissue sections were cut using Cryostat (Leica Microsystems, CM1950) and mount on SuperFrost Plus Adhesion slides (Erpedia, #J1800AMNZ). Dry slides were stored at –80°C.

For conventional light microscopy, tissue sections were fixed with acetone for 15 min and subjected to hematoxylin/eosin staining. DM6000-M microscope (Leica Microsystems) was used to acquire images.

For immunofluorescence microscopy, tissue sections were fixed with 4% PFA for 10 min, permeabilized with 0.1% Triton X-100 for 10 min, and blocked using PBS/5% FBS, 5% BSA for 1 hr. Next, samples were stained with the primary antibody mix (guinea pig anti-mouse insulin, rat anti-mouse CD4 Alexa Fluor 647, and rabbit anti-mouse CD8α) at 4°C overnight. After washing, the slides were incubated with the secondary antibodies for 1 hr, nuclei were stained with 5 μM DAPI solution (Invitrogen, D21490), and samples were mounted with ProLong Gold Antifade Mountant (Invitrogen, #P36930). Images were acquired using SP8 LIGHTNING confocal microscope (Leica Microsystems).

## ELISA

The anti-idiotypic B1 mAb were produced by the use of B1 hybridoma via the conventional ascites producing approach in paraffin oil pre-treated BALB/C mice. It was purified by 45% supercritical anti-solvent precipitation (ammonium sulfate) followed by extensive dialysis against distilled water, centrifuged at 12,000 × $g$ for removal of IgM, and purified by a protein A affinity chromatography. Next, it was biotinylated with Sulpho NHS-biotin reagent (Pierce) according to the manufacturer's protocol.

Blood of mice was taken on days 28, 54, 94 (or 95, or 96) of B-cell leukemia/lymphoma experiment. Blood serum was separated and serially diluted (1:40–1:640) in PBS. Plate wells (Costar) were coated with 50 μl of diluted samples and incubated overnight at 4°C, followed by blocking with 1% gelatin (200 μl per well, 2 hr at room temperature). Biotinylated anti-idiotypic B1 mAb was added (20 ng/ml) in a buffer containing 0.5% gelatin, 3% PEG, and 0.1% tween, and plate was incubated for 2 hr at room temperature. Next, samples were conjugated with ExtrAvidin−Peroxidase (Sigma-Aldrich) for 1 hr at room temperature, and 3,3′,5,5′-tetramethylbenzidine substrate (Sigma-Aldrich) was added for 10 min in dark. Reaction was stopped with 50 μl of 2 M $H_2SO_4$, and absorbance at 450 nm was measured by a Biolisa spectrometer (Bioclin).

## Bone marrow-derived dendritic cells

Bone marrow cells were seeded on 100 mm plates (tissue culture untreated) and maintained in DMEM (Sigma-Aldrich) containing 10% FBS (GIBCO), 100 U/ml penicillin (BB Pharma), 100 mg/ml streptomycin (Sigma-Aldrich), 40 mg/ml gentamicin (Sandoz), and 2% of supernatant from J558 cells producing GM-CSF for 10 days at 5% $CO_2$ at 37°C (*Kralova et al., 2018*). The cells were split every 2–3 days. On day 10, cells were incubated in the presence of 100 ng/ml LPS (Sigma-Aldrich) and 200 nM of indicated peptide for 3 hr at 5% $CO_2$ at 37°C. Next, plates were incubated with 0.02% EDTA in PBS for 5 min at 5% $CO_2$ at 37°C and harvested. Washed and filtered cells were used for adoptive transfers.

## In vitro proliferation assay

CD8[+] T lymphocytes from WT mice were FACS sorted, labeled with CTV, and plated into an anti-CD3ε-antibody-coated 48-well plate in IMDM (10% FBS [Gibco], 100 U/ml penicillin [BB Pharma], 100 mg/ml streptomycin [Sigma-Aldrich], 40 mg/ml gentamicin [Sandoz]). Tregs, sorted as CD4[+] GFP[+] T lymphocytes from DEREG[+] mice were added to corresponding wells of the plate in 1:1 ratio with

conventional CD8$^+$ T lymphocytes. Recombinant IL-2 (2 ng/ml) was added or not. Cells were incubated at 37°C, 5% $CO_2$ for 72 hr and analyzed by flow cytometry.

### In vivo proliferation assay

OT-I CD8$^+$ and OT-II CD4$^+$ T cells were isolated from OT-I $Rag2^{-/-}$ and OT-II $Rag2^{-/-}$ mice, respectively, using MACS negative selection kits. On day 0, mixture of OT-I CD8$^+$ (0.75 × 10$^6$ cells) and OT-II CD4$^+$ (1.5 × 10$^6$ cells) CFSE labeled cells was adoptively transferred into recipient Ly5.1 mice. The next day, recipient mice were immunized with ovalbumin protein i.p. (75 µg/mouse), and after 6 hr, they received the first dose of IL-2ic (1.5 µg IL-2 equivalent/dose, i.p.). On days 2–4, mice received additional IL-2ic doses. On day 5 post-immunization, spleens of Ly5.1 mice were collected and used for flow cytometry analysis.

### Treg depletion

In order to deplete Tregs, 0.25 µg of DT (#D0564, Sigma-Aldrich) was administered i.p. to DEREG$^+$ RIP.OVA mice and control DEREG$^-$ RIP.OVA mice for two consecutive days.

### Model of autoimmune diabetes

The model of autoimmune diabetes has been described previously (*King et al., 2012*). Briefly, an indicated number of OT-I T cells isolated from OT-I $Rag2^{-/-}$ mice were adoptively transferred into a recipient RIP.OVA mouse i.v. On the following day, the recipient mice were immunized with 10$^6$ of bone marrow-derived DCs loaded with indicated peptide (i.v.) or with 25 µg LPS + 50 µg OVA peptide in 200 µl PBS (i.p.). For the experiments using antigen-loaded DCs (*Figure 2B* and later), we included an internal control of two RIP.OVA mice (DEREG$^-$ and DEREG$^+$) receiving 1000 OT-I T cells and 10$^6$ OVA-loaded DCs in each experiment. When these control mice did not develop diabetes, which happened sometimes probably because of the issues with DC culture, the whole experiment was excluded. We realized that ~50% of mice die when the diabetic protocol is combined with IL-2/JES6 immunocomplexes (*Figure 4E–H*).

Alternatively, $Cd3e^{-/-}$ RIP.OVA recipient mice received 10$^6$ of polyclonal CD8$^+$ T lymphocytes derived from RIP.OVA Ly5.1 donors 8 days prior to immunization. One day prior to immunization, $Cd3e^{-/-}$ RIP.OVA mice received 4–8 × 10$^5$ Tregs (sorted as CD4$^+$ GFP$^+$ TCRβ$^+$) or 10$^6$ conventional CD4$^+$ T lymphocytes (sorted as CD4$^+$ GFP$^-$ TCRβ$^+$) derived from DEREG RIP.OVA donors.

If IL-2ic were used, 1.5 µg IL-2 equivalent/dose of IL-2/S4B6 or IL-2/JES6 were injected i.p. on days 0, 1, and 2 post-immunization. Alternatively, mice received five doses of IL-2/JES6 (2.5 µg IL-2 equivalent) on days −7,−6, −5,−4, −3 prior to the immunization. We removed these mice because they died before the first glucose measurement (a pre-established criterium). After two experiments with the same result, we did not repeat this assay anymore for ethical reasons.

Urine glucose was monitored on a daily basis using test strips (GLUKOPHAN, Erba Lachema). Blood glucose was measured using Contour blood glucose meter (Bayer) on a specified day(s), depending on the experimental design. The animal was considered diabetic when the concentration of glucose in the urine reached ≥1000 mg/dl for two consecutive days.

### Induction of KILR effector T cells using IL-2ic

On day 0, recipient mice (OT-I $Rag2^{-/-}$ or RIP.OVA) received 25 µg OVA peptide in 200 µl of PBS and/or 0.75 µg IL-2 equivalent/dose of IL-2/S4B6 in 250 µl PBS i.p. On days 1 and 2, mice received two more doses of IL-2/S4B6. On day 3, spleens were collected and used for flow cytometry analysis or FACS sort.

### Competitive adoptive transfer of KILR and control cells

OT-I $Rag2^{-/-}$ and Ly5.1 OT-I $Rag2^{-/-}$ mice were immunized with OVA peptide (25 µg, single dose on day 0, i.p.) and/or IL-2/S4B6 (0.75 µg IL-2 equivalent/dose of IL-2/S4B6, three doses, days 0–2, i.p.). On day 3, three populations of CD8α$^+$ T cells were FACS sorted: KILR (KLRK1$^+$ CD49d$^-$), naïve (KLRK1$^-$ CD49d$^-$), or effector cells (KLRK1$^-$ CD49d$^+$). KILR cells were mixed with naïve or effector cells (1:1, 500 × 10$^3$ cell/mouse in total in experiment 1, 400 × 10$^3$ cell/mouse in total in experiment 2), stained with CTV, and injected i.v. to recipient $Cd3e^{-/-}$ mice. On day 7, splenocytes of the recipient mice were analyzed by flow cytometry.

## In vivo killing assay

In vivo CD8$^+$ T cell killing assay was performed as described previously (*Kim et al., 2014*) with minor modifications. In short, OT-I *Rag2$^{-/-}$* mice were immunized or not with OVA peptide (25 µg, single dose on day 0, i.p.) and/or IL-2/S4B6 (0.75 µg IL-2 equivalent/dose of IL-2/S4B6, three doses, days 0–2, i.p.). On day 3, FACS sorted KLRK1$^+$ CD8α$^+$ or KLRK1$^-$ CD8α$^+$ cells were injected i.v. to recipient RIP. OVA mice, which had received a mixture of target cells (10$^7$ OVA-pulsed cells and 10$^7$ unpulsed target cells) earlier the same day. Target cells were prepared from spleens of Ly5.1 mice. OVA-pulsed cells were prepared via 1 hr incubation in RPMI-1640 containing 10% FBS (GIBCO), 100 U/ml penicillin (BB Pharma), 100 mg/ml streptomycin (Sigma-Aldrich), 40 mg/ml gentamicin (Sandoz), and 2 µM OVA peptide at 37°C, 5% CO$_2$, followed by loading with CTV. Unpulsed target cells were incubated in parallel in the medium without OVA peptide and subsequently loaded with CFSE. On day 4, splenocytes of the recipient mice were analyzed by flow cytometry. Target cells were identified as CD45.1$^+$ cells. Ratio of unpulsed (CFSE$^+$) to OVA pulsed target cells (CTV$^+$) was determined and normalized to those of recipients that did not receive OT-I T cells.

## Murine B-cell leukemia

On day 0, BALB/C female mice were injected i.p. with 5 × 10$^5$ BCL1 cells (a BALB/c-derived leukemia cell line) (*Slavin and Strober, 1978*) in PBS. On days 11 and 24 post-inoculation, mice received doxorubicin (Adriblastina) (5 mg/kg in 250 µl PBS, i.v.), followed or not by anti-CD4 or anti-CD8α depletion mAbs (200 µg in 250 µl of PBS for both, i.p.). On days 12, 13, 14, 25, 26, and 27 IL-2/JES6 was administrated to mice (5 µg IL-2 equivalent/dose in 250 µl PBS, i.p.). Survival of mice was monitored from day 30 to day 100. On days 28, 54, and 94–96 blood serum of mice was used for ELISA for antibody against the idiotype of IgM expressed on the BCL1 cells. For the flow cytometry analysis, spleens were harvested on day 30.

## B16F10 melanoma

Female C57Bl/6J mice were inoculated s.c. with 5 × 10$^5$ B16F10 melanoma cells (day 0). On day 6, mice received doxorubicin (Adriblastina) (8 mg/kg in 250 µl PBS, i.v.) followed or not by anti-CD8α or anti-CD4 depletion mAbs (200 ug in 250 µl of PBS, i.p.). On days 7–9, IL-2/JES6 was injected to mice (5 µg IL-2 equivalent/dose in 250 µl PBS, i.p.). Survival of mice was monitored on a daily basis.

Tumor size was measured as the width and length using caliper every 2–4 days. Length of the tumor was determined as the longest diameter. The width was determined as longest diameter of the tumor perpendicularly to the length. The thickness of the tumor was arbitrary assigned to be a half of the tumor width. Tumor volume (mm3) was calculated as V = (L × W × W)/2, where V is tumor volume, W is tumor width, and L is tumor length.

For the flow cytometry analysis, tumors and spleens were harvested on day 11. Tumor Dissociation Kit (Miltenyi Biotec, Cat# 130-096-730) has been used to prepare single-cell suspensions according to the manufacturer's protocol.

## ScRNA sequencing

Six-week-old female DEREG$^-$ RIP.OVA (n = 3) and DEREG$^+$ RIP.OVA (n = 3) littermate mice were treated with DT (0.250 µg per mouse in 0.5 ml of PBS, i.p.) on days –2 and –1 prior to the immunization, followed by i.v. injection of 5 × 10$^4$ OT-I T cells in 200 µl of PBS on day –1. OT-I T cells were obtained from spleens and lymph nodes of Ly5.1 OT-I *Rag2$^{-/-}$* mouse. On day 0, mice received Ly5.1 dendritic cells loaded with OVA (10$^6$ cells per mouse in 200 µl of PBS). On day 3 post-immunization, mice were sacrificed and spleens were collected. Erythrocytes were lysed with ACK buffer (2 min, on ice), cells were washed and resuspended in PBS/2% FBS. Next, cells were stained with LIVE/DEAD near-IR dye, anti-CD8a BV421, anti-CD45.1 PE, and anti-CD45.2 APC antibodies together with TotalSeq-C anti-mouse hashtag antibodies (anti-CD45 clone 30-F11, anti-H-2 clone M1/42, BioLegend, #155869 (MH5), #155871 (MH6), #155873 (MH7), #155875 (MH8), #155877 (MH9) and #155879 (MH10)). Viable CD8a$^+$, CD45.1$^+$, CD45.2$^-$ OT-I cells were FACS sorted. The individual samples were pooled together and with cells coming from an unrelated experiment. These unrelated cells were labeled with unique hashtag antibodies and removed during the analysis. The viability and concentration of cells after sort were measured using the TC20 Automated Cell Counter (#1450102, Bio-Rad). The viability of the cells pre-loading was >90%. Cells were loaded onto a 10X Chromium machine (10X Genomics)

aiming at the yield of 1000 cells per sample and processed with Feature Barcode technology for Cell Surface Protein protocol (#CG000186 Rev D) with the Chromium Single Cell 5' Library & Gel Bead and Chromium Single Cell 5' Feature Barcode Library kits (10X Genomics, #PN-1000014, #PN-1000020, #PN-1000080, #PN-1000009, #PN-1000084). Resulting cDNA libraries were sequenced on a NovaSeq 6000 (Illumina) with the S1 Reagent Kit (100 or 300 cycles, Illumina, #20012865, #20012863).

## Analysis of scRNAseq data

The raw scRNA data were mapped to Mouse Reference GRCm38 obtained from Ensembl database v102 (*Howe et al., 2021*) by 10X Genomics Cell Ranger 5.0.0 (*Zheng et al., 2017*). The same software was also used to create the employed mouse transcriptome reference. Default parameters were kept.

The hashtag sequences were mapped using the hashtag sequence references by 10X Genomics Cell Ranger 5.0.0. The data were pre-processed using Seurat R package v4.0.3 (*Haberman et al., 2014*) on R v4.0.4 (https://www.r-project.org/). The cells with less than 200 transcripts were removed. A histogram of cell counts having a specified number of hashtag reads was computed for each hashtag. To detect the end of an initial slope of a histogram, a function based on descent along the gradient – moving to the neighboring point in the histogram as long as its associated value is lower – was used. Before its application, each histogram was averaged using the sliding window of size n = 5 points. The resulting limit was used to associate or not each cell with the respective sample. Cells marked by multiple hashtags were considered as doublets and excluded from further analysis. Read limits (lim) and the number of recovered cells (#) for individual hashtags wer MH5 (lim 19, #744), MH6 (lim 19, #778), MH7 (lim 24, #789), MH8 (lim 21, #849), MH9 (lim 35, #836), and MH10 (lim 25, #878).

The lists of barcodes of each uniquely marked cell were created and used to separate read pairs where the start of first read R1 matches one of barcodes exactly from those with different barcodes and consequently either originating from the cells from unrelated experiment, from cell doublets or insufficiently marked cells. These reads were mapped again using 10X Genomics Cell Ranger 5.0.0 to generate data used further for downstream analysis. Cells with less than 200 transcripts and/or more than 15% of transcripts mapping to mitochondrial genes were removed. Mitochondrial genes, TCRα and TCRβ V(D)J-genes, ribosomal genes and genes whose transcripts were detected in less than three cells were excluded. Log normalization (scale factor = $1 \times 10^4$), scaling, identification of variable features (2000 variable features), dimensional reduction (PCA and UMAP with top 50 and 15 principal components, respectively, 40 nearest neighbors for UMAP), identification of nearest neighbors in the reduced space ('rann' algorithm), and Louvain clustering (resolution = 0.8) were performed using the Seurat R package v4.0.3 (*Haberman et al., 2014*) on R v4.0.4 (https://www.r-project.org/). These steps allowed to identify the low-quality cells and contaminating cell types that were removed along with cells with more than 7.5% of transcripts mapping to mitochondrial genes. In total, 4043 cells passed the QC steps. Afterward, all steps starting from and including log-normalization were repeated using slightly different parameters (800 variable features, 12 top principal components for both UMAP and PCA dimensional reductions and 30 nearest neighbors for UMAP, resolution = 0.3 for Louvain clustering; other parameters stayed the same). Cell cycle scores for S phase and G2/M phases were regressed out in the scaling step. Projection of clusters on a reference dataset of acute and chronic viral infection CD8+ T cells was done using the ProjecTILs R package v0.6.0 (*Andreatta et al., 2021*). NK cell signature was calculated for each cluster using the AddModuleScore function from the Seurat package with the default parameters. Signature genes were selected from the Molecular Signatures Database v7.5.1 (systematic name M4838 and M5669 for murine and human datasets, respectively). Signature genes that were not detected in our datasets were filtered out. For the separate analysis of KILR effector CD8+ T cells, cluster 4 was extracted from the original data and re-clustered again with adjusted parameters.

The code for the cell filtration to hashtags, barcode extraction, and whole downstream analysis is accessible on GitHub (https://github.com/Lab-of-Adaptive-Immunity/Supereffectors_scRNAseq; *Lab of Adaptive Immunity, 2023a*, copy archived at swh:1:rev:7b0dec9507dd45ab4bb0619912b240f51 2c7798f).

## Gene set enrichment analysis

Lists of IL-2-responsive genes were obtained from literature (*Kovanen et al., 2005*; *Lin et al., 2012*). Fold changes between DEREG+ and DEREG- samples were calculated with the Seurat R package

v4.0.3 (*Haberman et al., 2014*). Gene set enrichment analysis (GSEA) analysis was performed using the fgsea R package v1.16.0 (*Korotkevich, 2021*). Genes with similar fold changes were ranked in a random order.

## Building and analysis of human CD8+ T-cell atlas

The human CD8+ Atlas was built from 14 different data sets previously mapped to human genome reference GRCh38 by 10X Genomics and downloaded from their support site (https://support.10xgenomics.com/) either as raw feature matrices or, if not available for given data set, as filtered feature matrices. For data sets with available V(D)J information, their list was downloaded either as list of filtered annotated contigs or, if not available for given data set or it was originally processed by Cell Ranger 5.0.0 or higher, as list of all annotated contigs. The following data sets were used: *CD8+ T cells of Healthy Donor 1*, Single Cell Immune Profiling Dataset by Cell Ranger 3.0.2, 10X Genomics, (2019, May 9); *CD8+ T cells of Healthy Donor 2*, Single Cell Immune Profiling Dataset by Cell Ranger 3.0.2, 10X Genomics, (2019, May 9); *CD8+ T cells of Healthy Donor 3*, Single Cell Immune Profiling Dataset by Cell Ranger 3.0.2, 10X Genomics, (2019, May 9); *CD8+ T cells of Healthy Donor 4*, Single Cell Immune Profiling Dataset by Cell Ranger 3.0.2, 10X Genomics, (2019, May 9); *10k Human PBMCs with TotalSeq-B Human TBNK Antibody Cocktail, 3' v3.1*, Single Cell Gene Expression Dataset by Cell Ranger 6.0.0, 10X Genomics, (2021, March 31); *10k Human PBMCs Multiplexed, 2 CMOs - Inputs/Library*, Single Cell Gene Expression Dataset by Cell Ranger 6.0.0, 10X Genomics, (2021,March 2); *5k Peripheral blood mononuclear cells (PBMCs) from a healthy donor (v3 chemistry)*, Single Cell Gene Expression Dataset by Cell Ranger 3.0.2, 10X Genomics, (2019, May 29); *10k PBMCs from a Healthy Donor – Gene Expression and Cell Surface Protein*, Single Cell Gene Expression Dataset by Cell Ranger 3.0.0, 10X Genomics (2018, November 19); *8k PBMCs from a Healthy Donor*, Single Cell Gene Expression Dataset by Cell Ranger 2.1.0, 10X Genomics, (2017, November 8); *PBMCs of a Healthy Donor (v1)*, Single Cell Immune Profiling Dataset by Cell Ranger 3.1.0, 10X Genomics (2019, July 24); *Human T cells from a Healthy Donor, 1k cells – multi (v2)*, Single Cell Immune Profiling Dataset by Cell Ranger 5.0.0, 10X Genomics (2020, November 19); *Human PBMC from a Healthy Donor, 10k cells – multi (v2)*, Single Cell Immune Profiling Dataset by Cell Ranger 5.0.0, 10X Genomics, (2020, November 19); *Human PBMC from a Healthy Donor, 1k cells (v2)*, Single Cell Immune Profiling Dataset by Cell Ranger 4.0.0, 10X Genomics (2020, August 25); and *PBMCs of a healthy donor – 5' gene expression and cell surface protein*, Single Cell Immune Profiling Dataset by Cell Ranger 3.0.0, 10X Genomics, (2018, November 19).

Cells with less than 200 transcripts, above-average transcript count plus identified as doublets by scds R package v1.4.0 (*Bais and Kostka, 2020*) on R v4.0.4 (https://www.r-project.org/), and/or cells with more than 10% transcripts mapping to mitochondrial genes were removed. The genes equivalent to those removed in our mouse data set plus V(D)J genes of TCRγ and TCRδ genes were excluded. Centered log-ratio normalization of cell surface protein counts for data sets that have them, log-normalization (scale factor = $1 \times 10^4$) and identification of variable features for each data set (2500 variable features) were performed using Seurat R package v4.0.0 (*Haberman et al., 2014*). The same package was used for the integration of all data sets and subsequent scaling, dimensional reduction (PCA and UMAP with top 20 principal components), nearest neighbors identification and Louvain clustering (resolution = 0.4) of the resulting data set. The newly emerging cluster of cells with below-average gene count and above-average proportion of transcripts mapping to mitochondrial genes was removed. In total, 140,564 cells passed the QC steps. Afterward, the previous steps starting from and including both normalizations were repeated using the same parameters. A cluster containing 5160 MAIT cells was identified using differential expression analysis and V(D)J information, which was kept for some analyses, but removed for others. OPTICs method from dbscan R package v1.1–6 (*Hahsler, 2019*) was applied (parameters minPts = 500 and eps_cl = .55 for data both with and without MAIT cells) to generate the final clustering.

The code for building the atlas and its whole analysis is available on GitHub (https://github.com/Lab-of-Adaptive-Immunity/HS-CD8-Atlas; *Lab of Adaptive Immunity, 2023b*, copy archived at swh:1:rev:9b31b54fff516eba2a3ddb66449eb100db16521b).

## Statistical analysis

The number of biological replicates (mice) is shown in the respective figure legends. The data are pooled from two or more independent experiments. Statistical analysis was performed using an

appropriate test for given type of data using GraphPad Prism 5.0 or R. Quantitative data from mice were usually tested using nonparametric Mann–Whitney test or, for a comparison of more than two sample groups, Kruskal–Wallis test with Dunn's posttest, if required. Unpaired Student's $t$ test was used only once for the statistical analysis of the size of cell clusters from the scRNAseq experiments, where the limited number of biological replicates (n = 3) did not allow a nonparametric test with reasonable statistical power. However, the results were subsequently confirmed using independent assays with more biological replicates. Mann–Whitney test with Bonferroni correction (adjusted p-value) for multiple comparisons was used for the statistical analysis of differentially expressed genes in the scRNAseq data. The survival/disease-free curves were analyzed by log-rank (Mantel–Cox) test. The statistical test is indicated for each experiment in figure legends. All tests were two-tailed. When applicable, the correction for multiple comparisons was applied.

## Acknowledgements

We thank Ladislav Cupak for technical assistance and mouse genotyping. We thank members of the flow cytometry and animal facilities for their assistance. This project has received funding from the European Union's Horizon 2020 research and innovation programme under grant agreement no 802878 (ERC Starting Grant FunDiT to OS), National Institute of virology and bacteriology (Programme EXCELES, ID421 Project No. LX22NPO5103), funded by the European Union – Next Generation EU (OS), the Czech Science Foundation (19-03435Y to OS and 22-20548S to MK), Purkyne Fellowship of the Czech Academy of Sciences (to OS), Research Fund for Young Scientists at the University of Basel (DMS2336 to OS), the project National Institute for Cancer Research (Programme EXCELES, ID Project No. LX22NPO5102) – funded by the European Union-Next Generation EU (MK), the Institute of Molecular Genetics of the Czech Academy of Sciences (RVO 68378050 to OS), the Institute of Microbiology of the Czech Academy of Sciences (RVO 61388971 to MK), and Charles University Grant Agency (1706119 to OT and TC). The animal facility of the Institute of Molecular Genetics is a part of the Czech Center for Phenogenomics supported by the Czech Ministry of Education, Youth and Sports and the European Regional Development Fund (LM2015040, LM2018126, OP RDI CZ.1.05/2.1.00/19.0395, OP RDI BIOCEV CZ.1.05/1.1.00/02.0109).

VN and SJ are students of the Faculty of Science, Charles University in Prague.

## Additional information

### Funding

| Funder | Grant reference number | Author |
| --- | --- | --- |
| European Research Council | FunDiT | Ondrej Stepanek |
| European Union - Next Generation EU | LX22NPO5103 | Ondrej Stepanek |
| European Union - Next Generation EU | LX22NPO5102 | Marek Kovar |
| Czech Science Foundation | 19-03435Y | Ondrej Stepanek |
| Czech Science Foundation | 22-20548S | Marek Kovar |
| Research Fund for Young Scientists at the University of Basel | DMS2336 | Ondrej Stepanek |
| Charles University Grant Agency | 1706119 | Oksana Tsyklauri |
| Czech Science Foundation | 22-18046S | Ondrej Stepanek |

The funders had no role in study design, data collection and interpretation, or the decision to submit the work for publication.

## Author contributions
Oksana Tsyklauri, Formal analysis, Validation, Investigation, Visualization, Methodology, Writing – original draft, Writing – review and editing; Tereza Chadimova, Formal analysis, Investigation, Methodology, Writing – review and editing; Veronika Niederlova, Data curation, Software, Formal analysis, Validation, Visualization, Methodology, Writing – original draft, Writing – review and editing; Jirina Kovarova, Iva Malatova, Sarka Janusova, Helene Rossez, Hana Vecerova, Virginie Galati, Formal analysis, Investigation, Writing – review and editing; Juraj Michalik, Data curation, Software, Formal analysis, Visualization, Writing – original draft, Writing – review and editing; Olha Ivashchenko, Investigation, Methodology; Ales Drobek, Formal analysis, Investigation, Project administration, Writing – review and editing; Marek Kovar, Formal analysis, Supervision, Funding acquisition, Investigation, Project administration, Writing – review and editing; Ondrej Stepanek, Conceptualization, Data curation, Formal analysis, Supervision, Funding acquisition, Investigation, Visualization, Methodology, Writing – original draft, Project administration, Writing – review and editing

## Author ORCIDs
Oksana Tsyklauri ⓘ http://orcid.org/0000-0001-9997-5913
Veronika Niederlova ⓘ http://orcid.org/0000-0001-8768-9039
Juraj Michalik ⓘ http://orcid.org/0000-0001-8479-0991
Sarka Janusova ⓘ http://orcid.org/0000-0002-0111-497X
Olha Ivashchenko ⓘ http://orcid.org/0000-0002-5611-0933
Marek Kovar ⓘ http://orcid.org/0000-0002-6602-1678
Ondrej Stepanek ⓘ http://orcid.org/0000-0002-2735-3311

## Ethics
Animal protocols were performed in accordance with the laws of the Czech Republic and Cantonal and Federal laws of Switzerland, and approved by the Czech Academy of Sciences (identification no. 11/2016, 81/2018, 15/2019) or the Cantonal Veterinary Office of Baselstadt, Switzerland, respectively.

## Decision letter and Author response
Decision letter https://doi.org/10.7554/eLife.79342.sa1
Author response https://doi.org/10.7554/eLife.79342.sa2

# Additional files

## Supplementary files
• Supplementary file 1. Differentially expressed genes for all clusters shown in *Figure 5B*. The used differential expression criteria were fold change above 2, minimum of 0.1 difference in the fraction of detection between each cluster and the rest of the cells and adjusted p-value<0.01. For each gene, the fractions of cells with at least one detected transcript in the tested cluster (pct.1) or among all other cells (pct.2) are listed. The columns 'p_val', 'avg_log2FC' and 'p_val_adj' show p-values (Mann–Whitney $U$ test), log2 fold changes and adjusted p-values (Bonferroni correction).

• Supplementary file 2. Differentially expressed genes between the super-effector-like cell cluster and all the remaining cells in the human CD8$^+$ atlas (after MAIT removal). The used differential expression criteria were average fold change > 1.5 and adjusted p-value<0.01. For each gene, the fractions of super-effector-like cells (pct.1) or remaining cells (pct.2) with at least one detected transcript are listed. The columns 'p_val ', 'avg_log2FC' and 'p_val_adj' show p-values (Mann–Whitney U test), log2 fold changes and adjusted p-values (Bonferroni correction).

• MDAR checklist

## Data availability
All scRNA data analyzed in this study as well as the scripts used for the analysis are available without restrictions. The scRNAseq data generated in this study were deposited in the Gene Expression Omnibus (GSE183940).

The following dataset was generated:

| Author(s) | Year | Dataset title | Dataset URL | Database and Identifier |
|-----------|------|---------------|-------------|------------------------|
| Tsyklauri O, Chadimova T, Niederlova V, Michalik J, Janusova S, Rossez H, Drobek A, Vecerova H, Galanti V, Kovar M, Stepanek O | 2022 | Regulatory T cells suppress the formation of super-effector CD8 T cells by limiting IL-2 | https://www.ncbi.nlm.nih.gov/geo/query/acc.cgi?acc=GSE183940 | NCBI Gene Expression Omnibus, GSE183940 |

The following previously published datasets were used:

| Author(s) | Year | Dataset title | Dataset URL | Database and Identifier |
|-----------|------|---------------|-------------|------------------------|
| 10x Genomics | 2019 | CD8+ T cells of Healthy Donor 1 | https://www.10xgenomics.com/resources/datasets/cd-8-plus-t-cells-of-healthy-donor-1-1-standard-3-0-2 | 10xGenomics, cd-8-plus-t-cells-of-healthy-donor-1-1-standard-3-0-2 |
| 10x Genomics | 2019 | CD8+ T cells of Healthy Donor 2 | https://www.10xgenomics.com/resources/datasets/cd-8-plus-t-cells-of-healthy-donor-2-1-standard-3-0-2 | 10xGenomics, cd-8-plus-t-cells-of-healthy-donor-2-1-standard-3-0-2 |
| 10x Genomics | 2019 | CD8+ T cells of Healthy Donor 3 | https://www.10xgenomics.com/resources/datasets/cd-8-plus-t-cells-of-healthy-donor-3-1-standard-3-0-2 | 10xGenomics, cd-8-plus-t-cells-of-healthy-donor-3-1-standard-3-0-2 |
| 10x Genomics | 2019 | CD8+ T cells of Healthy Donor 4 | https://www.10xgenomics.com/resources/datasets/cd-8-plus-t-cells-of-healthy-donor-4-1-standard-3-0-2 | 10xGenomics, cd-8-plus-t-cells-of-healthy-donor-4-1-standard-3-0-2 |
| 10x Genomics | 2021 | 10k Human PBMCs with TotalSeq-B Human TBNK Antibody Cocktail, 3' v3.1 | https://www.10xgenomics.com/resources/datasets/10-k-human-pbm-cs-with-total-seq-b-human-tbnk-antibody-cocktail-3-v-3-1-3-1-standard-6-0-0 | 10xGenomics, 10-k-human-pbm-cs-with-total-seq-b-human-tbnk-antibody-cocktail-3-v-3-1-3-1-standard-6-0-0 |
| 10x Genomics | 2021 | 10k Human PBMCs Multiplexed, 2 CMOs - Inputs/Library | https://www.10xgenomics.com/resources/datasets/10-k-human-pbm-cs-multiplexed-2-cm-os-3-1-standard-6-0-0 | 10xGenomics, 10-k-human-pbm-cs-multiplexed-2-cm-os-3-1-standard-6-0-0 |
| 10x Genomics | 2019 | 5k Peripheral blood mononuclear cells (PBMCs) from a healthy donor (v3 chemistry) | https://www.10xgenomics.com/resources/datasets/5-k-peripheral-blood-mononuclear-cells-pbm-cs-from-a-healthy-donor-v-3-chemistry-3-1-standard-3-0-2 | 10xGenomics, 5-k-peripheral-blood-mononuclear-cells-pbm-cs-from-a-healthy-donor-v-3-chemistry-3-1-standard-3-0-2 |

*Continued on next page*

*Continued*

| Author(s) | Year | Dataset title | Dataset URL | Database and Identifier |
|---|---|---|---|---|
| 10x Genomics | 2018 | 10k PBMCs from a Healthy Donor - Gene Expression with a Panel of TotalSeq-B Antibodies | https://www.10xgenomics.com/resources/datasets/10-k-pbm-cs-from-a-healthy-donor-gene-expression-and-cell-surface-protein-3-standard-3-0-0 | 10xGenomics, 10-k-pbm-cs-from-a-healthy-donor-gene-expression-and-cell-surface-protein-3-standard-3-0-0 |
| 10x Genomics | 2017 | 8k PBMCs from a Healthy Donor | https://www.10xgenomics.com/resources/datasets/8-k-pbm-cs-from-a-healthy-donor-2-standard-2-1-0 | 10xGenomics, 8-k-pbm-cs-from-a-healthy-donor-2-standard-2-1-0 |
| 10x Genomics | 2019 | PBMCs of a Healthy Donor (v1) | https://www.10xgenomics.com/resources/datasets/pbm-cs-of-a-healthy-donor-v-1-1-1-standard-3-1-0 | 10xGenomics, pbm-cs-of-a-healthy-donor-v-1-1-1-standard-3-1-0 |
| 10x Genomics | 2020 | Human T cells from a Healthy Donor, 1k cells - multi (v2) | https://www.10xgenomics.com/resources/datasets/human-t-cells-from-a-healthy-donor-1-k-cells-multi-v-2-2-standard-5-0-0 | 10xGenomics, human-t-cells-from-a-healthy-donor-1-k-cells-multi-v-2-2-standard-5-0-0 |
| 10x Genomics | 2020 | Human PBMC from a Healthy Donor, 10k cells - multi (v2) | https://www.10xgenomics.com/resources/datasets/human-pbmc-from-a-healthy-donor-10-k-cells-multi-v-2-2-standard-5-0-0 | 10xGenomics, human-pbmc-from-a-healthy-donor-10-k-cells-multi-v-2-2-standard-5-0-0 |
| 10x Genomics | 2020 | Human PBMC from a Healthy Donor, 1k cells (v2) | https://www.10xgenomics.com/resources/datasets/human-pbmc-from-a-healthy-donor-1-k-cells-v-2-2-standard-4-0-0 | 10xGenomics, human-pbmc-from-a-healthy-donor-1-k-cells-v-2-2-standard-4-0-0 |
| 10x Genomics | 2018 | PBMCs of a Healthy Donor - 5' Gene Expression with a Panel of TotalSeq-C Antibodies | https://www.10xgenomics.com/resources/datasets/pbm-cs-of-a-healthy-donor-5-gene-expression-and-cell-surface-protein-1-standard-3-0-0 | 10xGenomics, pbm-cs-of-a-healthy-donor-5-gene-expression-and-cell-surface-protein-1-standard-3-0-0 |

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
