## [Editor Report]

This important study extends our understanding of how regulatory T cells can modulate the function of effector CD8 T cells. The evidence supporting the claims of the authors is solid and direct test the proposed mode of action. The work will be of interest to immunologists working on immune cell regulation.

---

## [Decision Letter]

**Decision letter after peer review:**

Thank you for submitting your article "Regulatory T cells suppress the formation of potent KLRK1 and IL-7R expressing effector CD8 T cells by limiting IL-2" for consideration by *eLife*. Your article has been reviewed by 2 peer reviewers, and the evaluation has been overseen by a Reviewing Editor and Tadatsugu Taniguchi as the Senior Editor. The reviewers have opted to remain anonymous.

Essential revisions:

1) As you will note from the individual reviews, there are key essential revisions required, in particular to the 2 points raised by Rev.1.

*Reviewer #1 (Recommendations for the authors):*

a) In the introduction 3 references are provided for IPEX but none for the inhibitory role of Tregs in tumors. One reference supporting the latter claim should be added.

b) In the text references to supplementary material is not uniform. Sometimes it is referred to as Figure S1 and sometimes as Figure EV3.

c) Figure 4B: Was pSTAT5 measured in directly ex vivo OT-I T cells or were they incubated with IL-2?

*Reviewer #2 (Recommendations for the authors):*

1) Data interpretation and results

a) Blood glucose should be measured on different days, including day 0 and some other days. The blood glucose curve should be evaluated for the diabetes severity.

b) Authors also need to add the tumor volume curve to evaluate the treatment schedule in Figure 7.

---

## [Author Response]

Reviewer #1 (Recommendations for the authors):a) In the introduction 3 references are provided for IPEX but none for the inhibitory role of Tregs in tumors. One reference supporting the latter claim should be added.

We added a reference for inhibitory role of CD4^+^ Tregs in cancer (Togashi, Y., Shitara, K. and Nishikawa, H. Nat Rev Clin Oncol 16, 356–371 (2019)) in the introduction.

b) In the text references to supplementary material is not uniform. Sometimes it is referred to as Figure S1 and sometimes as Figure EV3.

We have changed and unified the formatting according to the journal guidelines.

c) Figure 4B: Was pSTAT5 measured in directly ex vivo OT-I T cells or were they incubated with IL-2?

The pSTAT5 was measured directly ex vivo (the cells were fixed immediately after the isolation). We emphasized this in the Methods of the revised version of the manuscript.

Reviewer #2 (Recommendations for the authors):1) Data interpretation and resultsa) Blood glucose should be measured on different days, including day 0 and some other days. The blood glucose curve should be evaluated for the diabetes severity.

We routinely measure blood in the urine on a daily basis and in the blood only once during the course of the experiment (Palmer et al. 2016, PMID: 27188212, Drobek et al. 2018, PMID: 29752423). The reason was that we did not stress the animals too much and that we know from 23 experiments and 427 mice that there is near perfect correlation between the urine and blood glucose (Author response image 1). The blood glucose as well as urine glucose levels are largely digital. In most cases the urine glucose is 0 or >1000 mg/dl. Accordingly, blood glucose is usually <12 mmol/l in healthy animals and >20 mmol/l in diabetic animals.

**Author response image 1. sa2fig1:** 

However, we repeated the most common experiment using 10,000 transferred OT-I T cells into DEREG- and DEREG+ RIP.OVA mice with the daily monitoring of blood glucose levels to the revised version of the manuscript (Fig. S1A). It shows that the increase of blood glucose is very sudden and “digital” as the urine glucose.

b) Authors also need to add the tumor volume curve to evaluate the treatment schedule in Figure 7.

We are adding the B16F10 tumor size curves (Fig. S7D). The problem is that the mice start to drop off relatively early in the untreated or Dox only controls (see the survival curves in Fig. 7E, Fig. S7C). Thus, the growth curve would be heavily influenced by the removal of mice with the largest tumors. There were also some technical issues. Although we measured the tumor sizes during the course of the experiment, we did not get matched data, so we could not analyze the growth of individual tumors. Moreover, we measured the tumor sizes on slightly different days in each of the two independent experiments. We admit that the experimental design was not perfect, but we decided not to repeat these experiments for ethical reasons.

We did our best to address this comment by showing two datasets – first, we are showing the tumor size (Fig. S7D) on day 13 or 14, when there were the majority of mice are still in the experiment (this is from the previous experiment shown in Fig. 7E). This shows clear differences between the tumor sizes among the experimental groups, which correlates with the survival. Second, we are showing the tumor growth up to day 11 (Fig. S7E) from the new experiments, when the mice were sacrificed for the analysis of splenic and tumor-infiltrating cells (Fig. 7F-H). These data show similar tumor sizes in the experimental groups before and during the treatment. It also shows differences in the tumor size between untreated control and the DOX or DOX/IL-2 treated mice after the treatment (Fig. S7D). The difference between DOX and DOX/IL-2 starts to appear only on day 11, although it is still small and not significant at this time point.